# Adaptive Important Region Selection with Reinforced Hierarchical Search for Dense Object Detection

**Dingrong Wang**
Rochester Institute of Technology
Rochester, NY 14623
dw7445@rit.edu

**Hitesh Sapkota** [†]
Amazon Inc.
Sunnyvale, CA 94089
sapkoh@amazon.com

**Qi Yu**[*]
Rochester Institute of Technology
Rochester, NY 14623
qi.yu@rit.edu

## Abstract

Existing state-of-the-art dense object detection techniques tend to produce a large number of false positive detections on difficult images with complex scenes because they focus on ensuring a high recall. To improve the detection accuracy, we propose an Adaptive Important Region Selection (AIRS) framework guided by Evidential Q-learning coupled with a uniquely designed reward function. Inspired by human visual attention, our detection model conducts object search in a top-down, hierarchical fashion. It starts from the top of the hierarchy with the coarsest granularity and then identifies the potential patches likely to contain objects of interest. It then discards non-informative patches and progressively moves downward on the selected ones for a fine-grained search. The proposed evidential Q-learning systematically encodes epistemic uncertainty in its evidential-Q value to encourage the exploration of unknown patches, especially in the early phase of model training. In this way, the proposed model dynamically balances exploration-exploitation to cover both highly valuable and informative patches. Theoretical analysis and extensive experiments on multiple datasets demonstrate that our proposed framework outperforms the SOTA models.

## 1 Introduction

Dense object detection enjoys a wide range of applications in diverse domains [21, 40]. Representative use cases include surveillance video tracking by the police and merchandise recognition for online shopping [42, 6]. Despite its importance, dense object detection is an inherently challenging task as it requires predicting the bounding boxes for all objects present in a given image irrespective of their shape, size, and number. The inborn complexity of images, such as shadow/occlusion, image size, shape, color, and texture could also pose a significant hindrance in the detection process resulting in a lower accuracy [15].

Existing efforts have contributed different techniques to address the key challenges in dense object detection. For instance, two-stage approaches have been popularized where the first stage extracts candidate objects and the second stage classifies the extracted object while providing the bounding boxes through a regression network [15]. Representative two-stage detectors include R-CNN [15] and Faster R-CNN [33]. Two-stage detectors are limited in the number of candidate object found in first stage through the regional proposal network (RPN) and suffer from a low recall, especially in dense scenarios. To tackle this, one-stage approaches have been explored that achieve a higher recall along with faster training and inference [26, 38]. There are mainly two types of one-stage approaches: Anchor-based (*e.g.,* RetinaNet [26]) and Anchor-free (*e.g.,* FCOS [38]). The former computes the bounding box of an object by regressing the offsets from a predefined anchor box whereas the latter directly outputs the position and size of an object. However, existing one-stage

---

[*]Corresponding author, [†] Work was completed during the PhD study at RIT, which is not related to the position at Amazon.

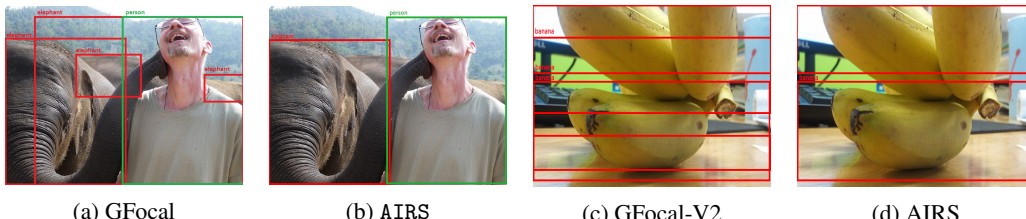

| (a) GFocal | (b) AIRS | (c) GFocal-V2 | (d) AIRS |

Figure 1: Bounding boxes produced by GFocal [23], GFocal-V2 [22], and AIRS, where GFocal, GFocal-V2 still tend to generate unnecessary bounding boxes resulting from false positive anchors, comparing to the proposed AIRS model.

detectors usually exhibit inconsistency in localization quality estimation between training and testing, stemming from the lack of supervision and resulting in many false positive anchors as shown in [23]. In testing, some negative anchors may generate an unusually high-quality estimation score and be selected as positive anchors (*i.e.,* false positives) due to lack of supervision. GFocal [23] alleviates this issue by leveraging the Focal Loss and aligning the localization quality and classification branches into a unified representation. While it achieved improved performance compared to previous one-stage detectors, GFocal still suffers from the problem of generating too many false positive predictions on small objects in a complex background because of selected low quality positive anchors, as shown in 1 (a) and (c).

Our analysis reveals that the non-adaptive criterion (which favors a high recall) in existing one-stage detectors does not capture diverse types of candidate anchors residing on the Feature Pyramid Network (FPN) [25] and will result into many false positive anchors. This phenomenon becomes prominent when testing images are difficult with complex/noisy background. To tackle this challenge, we propose to conduct **A**daptive **I**mportant **R**egion **S**election (AIRS) that is guided by a reinforcement learning (RL) agent performing *Evidential Q-learning* with a uniquely designed reward function. Similar to human visual attention, AIRS conducts object search in a top-down, hierarchical fashion. Benefiting from the top-down paradigm, the top layer in the hierarchy with a coarser granularity helps the model quickly identify most interesting regions that likely contain objects of interest. Only within those regions, the model performs a fine-grained search to more precisely locate the objects. Intuitively, the model only searches the patches from lower levels if the RL agent collects sufficient evidence on higher level supporting the presence of a potentially valuable object according to the learned evidential Q-value.

Furthermore, in the early phase of RL agent training, AIRS also encourages the agent to explore highly uncertain patches by leveraging the epistemic uncertainty provided by our evidential Q-value. Exploration of novel patches is also dynamically balanced with the exploitation of predicted high quality region. As a result, AIRS ensures that all potential patches have been adequately covered during the search process while avoiding the attendance of low-quality patches from fine-grained layers, leading to much improved precision without sacrificing the recall. As can be seen from Figure 1 (b) and (d), AIRS is able to identify all objects without producing any false positive bounding boxes. In contrast, GFocal, as shown in Figure 1(a) and (c), produces many false positive patches by paying too much attention on low-level details to maintain a high recall. To assess the effectiveness of AIRS, we perform extensive experiments on multiple real-world datasets with complex objects/backgrounds. Furthermore, we conduct a thorough theoretical analysis to show the convergence guarantee of our proposed evidential Q-learning. Our contributions are summarized below:

- an adaptive hierarchical object detection paradigm supported by an RL agent to mimic human visual attention that performs searching in the top-down fashion,
- novel evidential Q-learning driven by a unique reward function, covering both potentially positive and highly uncertain patches through dynamically balancing exploitation and exploration,
- theoretical guarantee on the fast convergence of the proposed evidential Q-learning algorithm,
- SOTA object detection performance outperforming strong baselines on challenging datasets.

## 2   Related Work

**Object Detection.** In dense object detection [10], both two-stage and one-stage detectors have been investigated in existing literature. Representative techniques in the former group include R-CNN [15], Faster R-CNN [33], where the detection involves two stages. Considering the limitation of

slow training and inference in two-stage detectors, multiple one-stage detectors have been proposed, such as FCOS, ATSS and GFocal series [38, 45, 23, 22]. However, these methods still suffer from attending too many false positive anchors (region proposals) resulting from the architecture design and the training setting. To overcome this, we design an uncertainty-guided RL to perform hierarchical search that effectively reduces the false positive detections.

**Deep Reinforcement Learning for Object Detection.** DRL formulates object detection problem as a Markov-Decision Process (MDP). It tries to find the salient parts of an image that are more probable to contain a target object, and then further zoom into them [4, 8]. There are generally two different action settings for this MDP. In the first setting, a hierarchical method is proposed by [4], where the agent chooses to focus on one of the 5 sub-regions of the image (ie. top-left, top-right, bottom-left, bottom-right, center) at each time step. In the second setting, a dynamic method is proposed by [8], where the agent deforms a bounding box using simple transformation actions (horizontal moves, vertical moves, scale changes, and aspect ratio changes) at each step to find the specific location of an object in the image. However, the above methods only detect a fixed number of objects. To overcome this issue, Ba et al. introduce a deep recurrent attention model (RAM) to recognize multiple objects [3]. Further, Zhou et al. propose ReinforceNet, which performs region selection and refinement by integrating RL's action space with CNN based feature space [47]. Different from all these works, `AIRS` leverages the advanced Feature Pyramid Network structure and performs RL-driven hierarchical search guided by epistemic uncertainty with much improved detection performance.

**Uncertainty in Deep Learning.** There have been different approaches to quantify uncertainty in deep learning models. Sensoy et al. [36] propose an Evidential Deep Learning (EDL) network, which treats the predicted multi-class probability as a multinomial opinion as developed in subjective logic [18]. Malinin et al. [30] propose Prior Networks (PNs) that consider distributional mismatch to explicitly quantify the distributional uncertainty. Amini et al. [1] propose an evidential regression network that quantifies the aleatoric and epistemic uncertainty based on the hyper-parameters. Different from previous work, we propose novel evidential deep Q-learning through an evidential regression Q-network to quantify the epistemic uncertainty, which is used to explore regions that the RL-agent is less familiar with.

## 3   Methodology

In this section, we first present the overall object detection process. We then go through each of the key component in our framework. We conduct a theoretical analysis to show how the proposed `AIRS` leverages effective exploration and hierarchical search to ensure theoretically guaranteed performance. Finally, we describe the training and inference process and explain how `AIRS` could be used for real-world object detection problems.

### 3.1   The Overall Detection Process

The overall detection process is guided by an RL agent as shown in Figure 2a. Our RL environment consists of a Feature Pyramid Network (FPN) that projects an input image into different resolutions organized into a hierarchical structure. As an example, the FPN in Fig. 2a has three layers with P5 having the lowest resolution and P3 having the highest resolution (these numbers follow the original FPN [25]). Once a patch is selected by the RL agent, it is passed through the feature extractor followed by the recurrent neural network (RNN) to generate the state representation. The state representation then goes through the evidential Q-network, which formulates an evidential Normal-inverse Gaussain (NIG) distribution and outputs the Q-value estimate for each available action. Then by combining with the corresponding epistemic uncertainty in the Q-value estimate, we have the evidential Q-value which balances the estimated Q-value with the (lack of) knowledge of the RL agent on the chosen action. Thus, in the initial phase of RL training, there are more uncertain patches and the RL agent is expected to explore more actively. As training progresses, the agent should try to choose those actions leading to patches (regions) that are most valuable. Based on the evidential Q-value obtained by balancing epistemic uncertainty and estimated Q-value, the agent takes action, and then selects the next patch with the goal of maximizing the expected reward.

### 3.2   Description of Key Components

**State Generation.**    To generate a state, we rely on the patches produced by RL environment from one of the $L + 1$ layers in the FPN, where $L$ is the top layer and $0$ being the bottom one.

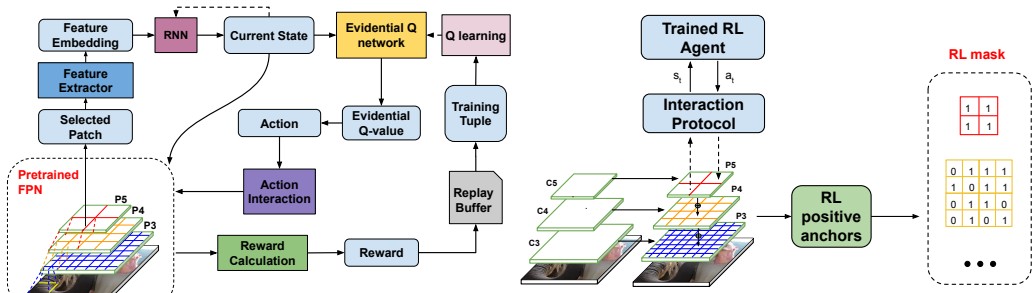

(a) Overview of the `AIRS` framework

(b) RL Inference to generate the binary RL mask

Figure 2: AIRS training and test pipelines

In the FPN, each patch in the higher layer (low resolution) is mapped to multiple patches in a lower layer (high resolution). Once a patch from layer $l$ is selected, it is passed through the feature extractor to get the feature embedding vector $\mathbf{e}_t \in \mathbb{R}^d$. Finally, the feature embedding and previous state representation ($\mathbf{s}_{t-1}$) are concatenated and passed through the RNN to generate the next state representation $\mathbf{s}_t = \text{RNN}(\mathbf{e}_t, \mathbf{s}_{t-1}; \boldsymbol{\theta}_r)$ where $\mathbf{s}_0 = \mathbf{0}$ and $\boldsymbol{\theta}_r$ is the parameters associated with the RNN. In this way, every state $\mathbf{s}_t$ captures the knowledge of previous observations from environment via $\mathbf{s}_{t-1}$.

**Evidential Q-Learning.** To ensure detection of all positive patches, it is crucial to perform effective exploration. To achieve this, we design evidential Q-learning that performs exploration in a systematic way by leveraging epistemic uncertainty. Let $D$ denote the size of the action space. The Q-value for $d^{th}$ action $q_{d,t}$ given state $\mathbf{s}_t$ is assumed to draw from a Gaussian distribution with mean $\mu_{d,t}$ and variance $\sigma_{d,t}^2$. We further place Gaussian and Inverse-Gamma priors on the mean and variance, respectively [5]:

$$q_{d,t} \sim \mathcal{N}(\cdot|\mu_{d,t}, \sigma_{d,t}^2), \ \mu_{d,t} \sim \mathcal{N}(\cdot|\gamma_{d,t}, \sigma_{d,t}^2 \nu_{d,t}^{-1}), \ \sigma_{d,t}^2 \sim \text{Inv-Gamma}(\cdot|\alpha_{d,t}, \beta_{d,t}) \tag{1}$$

where Inv-Gamma($\alpha_{d,t}, \beta_{d,t}$) is the Inverse Gamma distribution [1] and ($\gamma_{d,t}, \nu_{d,t}, \alpha_{d,t}, \beta_{d,t}$) are evidential Q-network outputs that form the evidential distributions as specified above. From these distributions, we can sample mean $\mu_{d,t}$ and variance $\sigma_{d,t}^2$ to generate Q-value estimate $q_{d,t}$. It should be noted that, because of the Inv-Gamma term, effective sampling for $q_{d,t}$ through the reparametrization trick [19] becomes difficult. We instead generate mean and variance with their expectations:

$$\mathbb{E}[\mu_{d,t}] = \gamma_{d,t}, \quad \mathbb{E}[\sigma_{d,t}^2] = \frac{\beta_{d,t}}{(\alpha_{d,t} - 1)} \tag{2}$$

Thus, the $q$-value is sampled as

$$q_{d,t} \sim \mathcal{N}\left(\cdot|\gamma_{d,t}, \frac{\beta_{d,t}}{(\alpha_{d,t} - 1)}\right) \tag{3}$$

Using this trick, the gradient could be easily traced back to evidential Q-network's hyper-parameter outputs ($\gamma_{d,t}, \nu_{d,t}, \alpha_{d,t}, \beta_{d,t}$) for each action $a_{d,t}$ from the corresponding Q-value $q_{d,t}$ in evidential Q-learning. In addition to the Q-value estimate, it is also essential to integrate uncertainty to facilitate exploration of unknown patches, leading to the following evidential Q-value

$$q_{d,t}^e = q_{d,t} + \lambda \text{Var}[\mu_{d,t}], \quad \text{Var}[\mu_{d,t}] = \frac{\mathbb{E}[\sigma_{d,t}^2]}{\nu_{d,t}} = \frac{\beta_{d,t}}{\nu_{d,t}(\alpha_{d,t} - 1)} \tag{4}$$

where $\text{Var}[\mu_{d,t}]$ captures the epistemic uncertainty and $\lambda$ balances epistemic uncertainty and Q-value. The evidential decomposition of the total uncertainty allows us to separate uncertainty caused by the noise in the data (*i.e.,* aleatoric uncertainty or $\mathbb{E}[\sigma_{d,t}^2]$) and uncertainty caused by lack of knowledge (*i.e.,* epistemic uncertainty or $\text{Var}[\mu_{d,t}]$). Since the evidential-Q value only integrates with the epistemic uncertainty, it ensures that the exploration will focus on improving the knowledge of the agent while being robust to the noise in the data.

To generate the action vector, we consider both $q_{d,t}^e$ and the constraints that avoid the agent moving into an invalid region that includes already visited patches and void space (*e.g.,* downward movements from layer 0, upward movement from layer $L$). To this end, we define a mask vector. Let

$m_{l,t}^d$ be the mask value (binary) associated with the $d^{th}$ action in $l^{th}$ layer in $t^{th}$ time step then the masked evidential Q-value is

$$\widetilde{\mathbf{q}_{d,t}^e} = \mathbf{q}_{d,t}^e \otimes \mathbf{m}_{l,t}^d \tag{5}$$

Let $j = \arg\max_d \{\widetilde{q_{d,t}^e}\}_{d=0}^{D-1}$, then the action value for each entry $d \in \{0, 1, .., D-1\}$ is updated:

$$a_{d,t} = \begin{cases} 1, \text{if } d = j; \\ 0, \text{otherwise} \end{cases} \tag{6}$$

More detailed information of the constraint mask design can be found in Appendix C.3.

Based on the masked evidential Q-value $\widetilde{\mathbf{q}_{d,t}^e}$, the RL agent selects the best action $a_t$ and receives a reward $r(\mathbf{s}_t, \mathbf{a}_t)$. The agent repeats the selection process until reaching a limit $T$ steps or arriving at the terminal condition (*i.e.,* upward movement in top-most layer $L$) is triggered. During each step, the agent stores the tuple of $\mathbf{s}_t, \mathbf{a}_t, r_t, \mathbf{s}_{t+1}$ into a replay buffer. After collecting $K$ training tuples, one batch of training tuples is sampled for off-policy Q-learning. The following loss is used to update the feature extractor ($\boldsymbol{\theta}_f$), RNN ($\boldsymbol{\theta}_r$), and evidential Q-network ($\boldsymbol{\theta}_e$):

$$\mathcal{L}_{\boldsymbol{\theta}_f, \boldsymbol{\theta}_r, \boldsymbol{\theta}_e} = (q_{d,t}(\boldsymbol{\theta}_f, \boldsymbol{\theta}_r, \boldsymbol{\theta}_e) - \widehat{q_{d,t}})^2, \; \widehat{q_{d,t}} = r(\mathbf{s}_t, \mathbf{a}_t) + \gamma \mathbb{E}_{s_{t+1} \sim D} \max_d (q_{d,t+1}) \tag{7}$$

where $\widehat{q_{d,t}}$ is evaluated using the Bellman equation.

**Action Interaction.** The action interaction module translates the action into the location of the next patch to be selected. It considers a $D$ dimensional action vector, where the first $D-1$ actions: $a_{d,t}, d \in [0, .., D-2]$ are downward movements that direct the agent currently located on layer $l$ into one of its mapping sub-patches in layer $l-1$ with $p_{i,t}^l$ being the current selected patch from $l$ layer. The location of the selected sub-patch from $l-1$ layer is provided by the index of the action-value whose entry is set to 1 (*e.g.,* $a_{0,t} = 1$ means top-left and $a_{1,t} = 1$ means top-right). The last action $a_{D-1,t}$ denotes the upward movement to the parent patch in layer $l+1$. The action interaction process is essentially a hierarchical tree search in FPN (with a virtual layer $L$ as the root node) and we provide illustrative examples in Appendix C.3.

**Reward Design.** The action space of the RL agent involves two major types of movements in the hierarchical search, downward and upward. To facilitate each type, we define a unique reward function. For **downward movement** actions, we compute the reward based on the ranking of the patch selected by the movement action compared to all other patches located on the same layer in terms of the number of the positive anchors they contain. Specifically, we compute the quality measure estimate of each anchor by investigating a range of metrics: centerness [38], IoU[32], GIoU[34], and DIoU [46]. It should be worth noting that all these metrics encode the supervised signal with a threshold to decide whether an anchor is positive or not. We follow the RetinaNet setting but use DIoU as our positive anchor criterion and conduct an ablation study to verify its superiority. After getting the positive anchors for each patch, we calculate the quality score $g(\mathbf{s}_t, \mathbf{a}_t)$ for each patch in terms of the number of positive anchors on it as "ground truth" information. The details can be seen in Appendix C.1. In addition, we set up a penalty term with the downward movement in each time step representing the searching cost. Such a cost should be related to both training progress and the depth of search. For example, searching a bottom layer's patch in a later training phase when the model gets enough knowledge of the input should be considered costly. Given this insight and by combining the above two factors, we design our unique reward

$$r(\mathbf{s}_t, \mathbf{a}_t) = g(\mathbf{s}_t, \mathbf{a}_t) - \frac{n_{epcoh}}{N_{epoch}} P_{s_t, a_t}^l \tag{8}$$

where $n_{epcoh}, N_{epoch}$ are the current and total training epoch, $P_{s_t, a_t}^l$ is the penalty term for each layer $l$, and $l$ is the layer index of next selected patch given $\mathbf{s}_t, \mathbf{a}_t$ in time step $t$. For **upward movement**, the reward is simply set to 0, which means when the downward movement's benefits for exploration cannot cover the search cost, the model prefers to return back and search other patches in the same layer. In this way, we achieve the exploitation-exploration balance in the reward design, besides the evidential Q-learning.

### 3.3 Theoretical Analysis

We establish the statistical guarantee for `AIRS` that integrates evidential Deep-Q learning with hierarchical search. Let $Q^*$ be the optimal action-value function and $Q^{\pi_k}$ be the action-value function corresponding to the policy $\pi_k$.

**Theorem 3.1.** *Under the assumption of the smoothness on the Bellman optimality operator, there exists a constant $C$ such that the following bound holds*

$$\|Q^* - Q^{\pi_K}\|_1 \leq C\phi_{e,f}\frac{\gamma}{1-\gamma^2}|\mathcal{A}|\tau(\eta,v) + \frac{4\gamma^{K+1}}{1-\gamma^2}R_{max} \tag{9}$$

*where $\mathcal{A}$ denotes the action space, $R_{max}$ upper bounds the uniquely designed reward, and $K$ is the total number of RL iterations. The term $\tau(\eta,v)$ is associated with the Hölder smoothness criterion on neural network function $f$, which is required here to ensure the finite sample guarantee with $\eta \in \mathbb{Z}$ denoting the upper limit of the number of input variables on which the Hölder smooth function depends. This integer essentially controls the statistical rate for estimating the Hölder smooth function [39]. Term $v$ is the exponent in the Hölder smooth function with $v = 1$ leading to the Lipschitz continuity. Finally, $\phi_{e,f}$ is the upper bound on the cumulative discounted concentration parameter over $K$ steps. It quantifies the similarity between the sampling distribution obtained through $f$ and the actual distribution of $K^{th}$ step Markov Decision Process (MDP) starting from the initial fixed distribution $e$ on initial state-action pairs $(\mathcal{S}_0, \mathcal{A}_0)$. A larger difference leads to a higher $\phi_{e,f}$.*

**Remarks.** The complete proof is given in the Appendix B along with a justification to ensure all the assumptions hold in our setting. First, the upper bound on the r.h.s. consists of two types of errors: statistical (first term) and algorithmic (second term), where the latter decreases exponentially because our reward is upper bounded. As a result, the bound is dominated by the former term after sufficient rounds of iterations. Second, the Hölder smoothness assumption on the Bellman optimality operator implies that the optimal action-value function $Q^*$ is close to the functional classes constituting the evidential Q-network and that functional classes are approximately closed under the Bellman operator. This completeness assumption ensures the finite sample guarantee for the proposed `AIRS`. In particular, the term $\tau(\eta,v) \propto n^{-f(\eta,v)}$ decays quickly with respect to the sample size $n$ making the bound even tighter. Third, thanks to the specially designed hierarchical search strategy, the action space in our setting is very small (*i.e.,* $D = 5$) making the $|\mathcal{A}|$ small, leading to a tighter bound. Finally, the $\phi_{e,f}$ term essentially provides the upper bound on the similarity between sampling distribution and actual distribution of the state-action space. By performing epistemic uncertainty guided exploration, our collected state-action samples are likely to be representative of the actual distribution. This makes $\phi_{e,f}$ small that further tightens the bound.

### 3.4 Training and Testing Procedures

The detailed training process is presented in Algorithm 1 and illustrated in 5 of Appendix C.2, where the search always starts from the top-most layer *i.e., $L$*. It should be noted that, we create the RL environment by leveraging the FPN structure of the pre-trained backbones (*i.e.,* ResNet-50). Based on the masked evidential Q-value estimate, the agent selects the next action, which would be either a downward or upward movement. Then, the agent moves to the next patch and continues the process until receiving an upward movement in layer $L$ or reaches the maximum time step *i.e., $T$*. After every $K$ such iterations, training tuples are sampled from the replay buffer for off-policy learning. For inference, we leverage the same pre-trained FPN as training, and run the trained RL agent on the test image's FPN to generate RL mask for its FPN structure, illustrated as Figure 2b. These binary RL masks are then used for patch selection in the test phase to generate more precise anchors from a large magnitude of candidate positive anchor pool to facilitate a more calibrated training (see Figure 4 of Appendix C.2). As can be seen, the training is more efficient comparing to other RL based methods, and the inference speed is also competitive w.r.t. the latest baselines (see Appendix D.6).

## 4 Experiments

We conduct extensive experimentation to evaluate the effectiveness of `AIRS`. We first describe three real-world object detection datasets. For each dataset, we also construct a subset of images with more complex scenes. These challenging subsets can provide direct evidence of reducing false positive predictions using `AIRS`. We then present the experimental setting consisting of evaluation metrics and experimental setup (such as network backbones used). We show the comparison results with competitive baselines in terms of object detection performance as well as inference cost in a quantitative study. Additionally, we conduct a qualitative analysis and ablation study to uncover deeper insights on performance advantage of `AIRS`. Finally, in Appendix D.4, we introduce the detailed training configurations to guarantee the success of our off-policy Q-learning.

### 4.1 Datasets

- *MS COCO* [27]: It contains 91 categories. Following [38, 26], we use the COCO trainval35k split containing 115K images for training and minival split containing 5K images for testing.
- *PASCAL Visual Object Classes (VOC) 2012* [12]: It contains 20 categories and is partitioned into three subsets: 5,717 images for training, 5,823 images for validation, and 10,991 images for testing.
- *Google Open Images V4* [20]: It contains 9M (million) image with 600 object categories, where training set contains 1.74 M images, validation set contains 125K images, and testing contains 41K images. It is worth noting that images in this dataset are very diverse and often contain complex scenes with several objects *i.e.,* on average 8.4 objects per image.
- *Challenging subset:* From each dataset, we construct a subset (denoted as 'CH') to include most challenging images using the following criteria: (a) images where the ratio of large and medium objects (area $\geq 32^2$) to small objects (area $< 32^2$) ranging from 1 to 1/2 to ensure smaller objects coexist and embedd within large objects making detection task much more challenging, (b) images where multiple objects overlap with each other, and (c) images where multiple small objects are embedded into the bigger one. Appendix D.7 show examples of selected images.

### 4.2 Experimental Settings

**Evaluation Metrics.** Following evaluation performed in the benchmark COCO dataset [27], we assess the performance using Average Precision (AP). Additionally, we separately report the AP performance for small, medium, and large objects named as $AP^S$, $AP^M$, $AP^L$ respectively. For the challenging subsets, we report the AP score named as $AP^{CH}$. It should be noted that small objects and constructed subsets correspond to the more challenging detection tasks and therefore we would expect a more significant performance gap compared with the baselines.

**Experimental setup.** For the FPN, we follow the same experiment setting as GFocal, which uses its pre-trained ResNet-50 as backbone, and applies a 3-layer FPN with patch size defined as a quarter of the area of the layer $L - 1$ (the top layer $L$ in our case is a virtual layer as the root node in the tree structure), but instead use DIoU as positive anchor criterion. For the feature extractor, we use a three-layer Multi-Layer Perceptron (MLP) structure. Through grid based hyper-parameter search using a validation dataset, we set the total training epochs $N_{epoch} = 12$, action space $D = 5$, maximum time step $T = 60$, discount factor $\gamma = 0.9$, learning rate = 0.001, and $\lambda = 1$. We gradually shrink $\lambda$ to balance exploitation and exploration. For the penalty term $P^l_{s_t,a_t}$ set up, we choose (0.3,0.6,0.9) for layers (P5, P4, P3) after hyper-parameter searching. For other baselines, we train them until convergence and test in the same data sets for fair comparison.

### 4.3 Quantitative Study

**Comparison baselines.** In our quantitative study, we include baselines that are most relevant to our model, including representative or latest two-stage detectors: Faster R-CNN [33], Cascade R-CNN [7], Reppoints [41], TridentNet [24], DETR [9], Co-DETR [49], EVA [14], and DINO [44], as well as most recent one stage detectors: RetinaNet [26], FCOS [38], ATSS [45], SAPD [48], SpineNet [11] and GFocal [23]. Faster R-CNN and Cascade R-CNN use an ROI pooling layer to extract candidate ROI regions first then regress from those regions, while Reppoints and TridentNet apply deformable convolution technique or scale-specific feature maps to handle scale and perspective variations in images. DETR leverages a set-based global loss that forces unique predictions via bipartite matching, and a transformer encoder-decoder architecture to effectively remove the need for Non maximal suppression and anchor generations. Co-DETR further applies a novel collaborative hybrid assignments training scheme on top of it. DINO improves DETR on de-noising the anchor boxes for end-to-end training. All one-stage methods use FPN, but the difference resides in the training loss (GFocal), and positive anchor criterion choices (RetinaNet, FCOS, ATSS, SAPD). Instead, EVA is a vanilla ViT pre-trained to reconstruct the masked out image-text aligned vision features conditioned on visible image patches for exploring the limits of visual representation at scale. For fair comparison, all baselines apply the same pre-trained backbone (*i.e.,* ResNet-50) to extract image features. For YOLO series comparison, since they are different from other one-stage detectors based on FPN, we separately compare with them in Appendix D.1.

**Comparison results.** Table 1 shows the performance in terms of AP for multiple datasets compared to competitive baselines. As shown, our approach has achieved better detection performance in general. Comparing to those RPN based two-stage frameworks which suffer from a low recall given limited candidate object predictions, our approach leverages abundant positive anchors provided by

Table 1: Detection performance comparison on all three datasets along with their challenging subsets

| Category | Method | MS COCO | | | | | Pascal VOC 2012 | | | | | Open Image V4 | | | | |
|---|---|---|---|---|---|---|---|---|---|---|---|---|---|---|---|---|
| | | AP | $AP^S$ | $AP^M$ | $AP^L$ | $AP^{CH}$ | AP | $AP^S$ | $AP^M$ | $AP^L$ | $AP^{CH}$ | AP | $AP^S$ | $AP^M$ | $AP^L$ | $AP^{CH}$ |
| Two-stage | Faster R-CNN [33] | 36.2 | 18.2 | 39.0 | 48.2 | 19.4 | 73.8 | 25.2 | 75.2 | 78.4 | 26.5 | 37.4 | 19.6 | 38.5 | 42.2 | 20.5 |
| | Cascade R-CNN [7] | 42.8 | 23.7 | 45.5 | 55.2 | 22.5 | 82.7 | 29.5 | 73.6 | 83.5 | 28.6 | 38.6 | 25.4 | 40.4 | 44.8 | 23.7 |
| | RepPoints [41] | 41.0 | 23.6 | 44.1 | 51.7 | 21.2 | 81.3 | 29.1 | 74.4 | 83.0 | 27.6 | 39.1 | 24.2 | 39.1 | 42.5 | 21.5 |
| | TridentNet [24] | 42.7 | 23.9 | 46.6 | 56.6 | 20.5 | 82.5 | 29.5 | 64.3 | 84.7 | 28.4 | 40.5 | 26.2 | 41.9 | 45.8 | 20.4 |
| | DETR [9] | 42.0 | 20.5 | 45.8 | 61.1 | 17.5 | 80.2 | 25.1 | 62.8 | 84.5 | 26.3 | 39.6 | 23.5 | 41.5 | 45.9 | 17.8 |
| | Co-DETR [49] | 42.5 | 20.8 | 46.2 | 61.5 | 17.9 | 80.5 | 25.4 | 63.2 | 84.9 | 26.5 | 39.7 | 23.9 | 41.8 | 46.3 | 18.3 |
| | EVA [14] | 46.7 | 28.5 | 48.2 | 61.9 | 28.8 | 84.7 | 31.5 | 75.4 | 86.5 | 28.7 | 44.1 | 25.8 | 46.5 | 50.8 | 26.7 |
| | DINO-4scale [44] | 47.8 | 30.2 | 50.1 | 62.3 | 29.0 | 86.9 | 33.4 | 77.2 | 88.5 | 30.9 | 46.2 | 29.8 | 47.8 | 52.3 | 28.1 |
| | DINO-5scale [44] | 47.9 | 30.0 | 50.4 | 62.5 | 29.0 | 87.1 | 33.3 | 77.4 | 88.6 | 31.2 | 46.4 | 29.9 | 47.7 | 52.4 | 28.2 |
| One-stage | RetinaNet [26] | 39.1 | 21.8 | 42.7 | 50.2 | 21.6 | 77.0 | 27.8 | 62.9 | 81.5 | 27.3 | 38.5 | 24.8 | 40.2 | 42.4 | 21.3 |
| | FCOS [38] | 41.5 | 24.4 | 44.8 | 51.6 | 23.5 | 83.3 | 31.4 | 64.2 | 85.8 | 30.5 | 40.3 | 26.1 | 41.8 | 45.4 | 23.2 |
| | ATSS [45] | 43.6 | 26.1 | 47.0 | 53.6 | 23.8 | 84.2 | 32.6 | 74.3 | 86.9 | 31.3 | 42.2 | 26.9 | 42.5 | 46.8 | 24.0 |
| | SAPD [48] | 43.5 | 24.9 | 46.8 | 54.6 | 22.4 | 83.8 | 31.5 | 75.3 | 86.2 | 29.5 | 41.1 | 25.9 | 41.6 | 45.8 | 23.5 |
| | SpineNet [11] | 41.5 | 23.3 | 45.0 | 58.0 | 21.2 | 82.6 | 29.3 | 73.5 | 85.7 | 27.4 | 40.2 | 25.8 | 41.2 | 45.3 | 21.6 |
| | GFocal [23] | 45.0 | 27.2 | 48.8 | 54.5 | 25.4 | 86.5 | 35.0 | 78.0 | 90.5 | 32.6 | 45.8 | 29.5 | 46.5 | 51.4 | 26.3 |
| Ours | AIRS | **48.3** | **32.1** | 48.5 | 54.3 | **29.4** | **88.7** | **37.3** | **79.0** | **91.5** | **35.6** | **47.5** | **31.5** | **48.1** | **53.1** | **29.0** |

Table 2: Detection performance using different backbone architectures

| Category | Method | MS COCO | | | | | Pascal VOC 2012 | | | | | Open Image V4 | | | | |
|---|---|---|---|---|---|---|---|---|---|---|---|---|---|---|---|---|
| | | AP | $AP^S$ | $AP^M$ | $AP^L$ | $AP^{CH}$ | AP | $AP^S$ | $AP^M$ | $AP^L$ | $AP^{CH}$ | AP | $AP^S$ | $AP^M$ | $AP^L$ | $AP^{CH}$ |
| Two-stage | DINO(Swin-T) [28] | 48.0 | 31.5 | **50.5** | **55.6** | 27.6 | 87.9 | 34.6 | 78.0 | 89.4 | 32.5 | 47.4 | 31.2 | 48.7 | 53.5 | 29.0 |
| | DINO(EfficientNet) [37] | 47.8 | 31.1 | 50.3 | 55.4 | 27.3 | 87.5 | 34.2 | 77.8 | 89.1 | 32.3 | 47.1 | 31.0 | 48.4 | 53.1 | 28.7 |
| | DINO(ConvNeXt) [29] | 48.1 | 31.7 | 50.4 | 55.5 | 27.7 | 88.2 | 34.7 | 78.1 | 89.6 | 32.8 | 47.5 | 31.5 | 48.7 | 53.2 | 29.2 |
| One-stage | GFocal (Swin-T) | 46.0 | 27.6 | 49.5 | 54.8 | 25.7 | 87.3 | 35.8 | 78.7 | 91.1 | 33.2 | 46.4 | 30.0 | 46.9 | 51.9 | 26.7 |
| | GFocal (EfficientNet) | 45.8 | 27.3 | 49.1 | 54.6 | 25.5 | 87.0 | 35.4 | 78.2 | 90.7 | 32.9 | 46.2 | 29.6 | 46.6 | 51.5 | 26.5 |
| | GFocal (ConvNeXt) | 46.2 | 27.7 | 49.8 | 55.0 | 25.8 | 87.5 | 36.0 | 78.9 | 91.4 | 33.5 | 46.6 | 30.3 | 47.1 | 52.2 | 27.0 |
| Ours | AIRS (Swin-T) | 48.9 | 32.8 | 49.1 | 54.9 | 29.8 | 89.8 | 38.4 | 79.8 | 92.8 | 36.5 | 48.5 | 32.3 | 49.0 | 53.9 | 30.4 |
| | AIRS (EfficientNet) | 48.7 | 32.6 | 48.8 | 54.5 | 29.5 | 89.4 | 38.1 | 79.8 | 92.6 | 36.1 | 48.3 | 32.1 | 48.6 | 53.6 | 30.2 |
| | AIRS (ConvNeXt) | **49.0** | **32.9** | 49.3 | 55.1 | **30.2** | **91.2** | **38.7** | **80.1** | **93.4** | **36.9** | **49.0** | **32.7** | **49.5** | **54.4** | **30.8** |

the underlying one-stage framework. Additionally, our approach can more effectively avoid false positive predictions benefiting from the learned RL masks that precisely detect positive anchors. This is clearly demonstrated through improved performance on small object detection (*i.e.,* $AP^S$) and the challenging subset (*i.e.,* $AP^{CH}$). As can be seen, the performance improvement compared to its base detector GFocal (without RL augmentation) is as high as around $4\%$ in certain cases. It is noted that the performance advantage is not as prominent in the medium and large object detection as these are relatively easy cases and can be adequately handled by commonly used models. Our proposed technique is designed to focus on difficult images while remaining competitive on easier detection tasks.

**Results on different backbones.** We also test our model's performance on multiple latest backbones such as Swin-T [28], Efficientnet-b3 [37] and ConvNeXt [29], and compare with the most competitive baselines of each category with the same backbone setting as shown in Table 2. The results show a highly consistent trend as in Table 1.

**Inference speed comparison.** Finally, we compare the parameter size as well as inference speed of our model with representative baselines. As the results shown in Table 3, the inference speed of AIRS does not bring extra detection burden compared to those latest baselines.

Table 3: Inference speed comparison

| Category | Model | Params | GFLOPS/FPS |
|---|---|---|---|
| Two-stage | Faster-RCNN | 40M | 194/21 |
| | DETR(DC5) | 41M | 216/25.4 |
| | Cascade R-CNN | 41M | 208/22.8 |
| | DINO-4scale | 47M | 212/24.1 |
| One-stage | FCOS | 32M | 165/19.4 |
| | ATSS | 32M | 168/19.4 |
| | SpineNet | 35M | 176/23.4 |
| | GFocal | 32M | 168/21.8 |
| Ours | AIRS | 32M | 165/20.7 |

Table 4: RL baseline comparison

| Model | mAP |
|---|---|
| PACNet [43] | 54.2 |
| Hierarchical-RL [4] | 46.1 |
| Caicedo-RL [8] | 28.1 |
| Tree-RL [17] | 73.1 |
| Multiple-RL [3] | 40.7 |
| ReinforceNet [47] | 53.4 |
| AIRS | **88.7** |

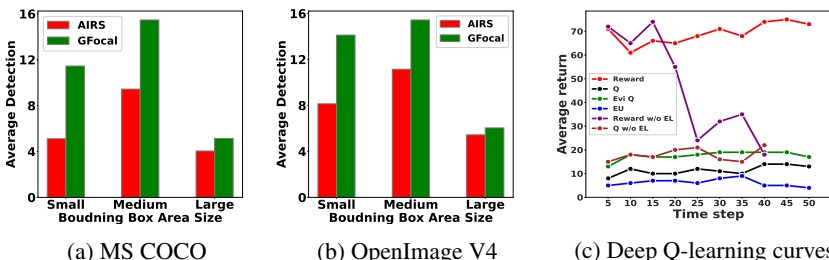

| (a) MS COCO | (b) OpenImage V4 | (c) Deep Q-learning curves |
|:---:|:---:|:---:|

Figure 3: (a)-(b) Average number of detections per test image based on the bounding box area on MS COCO and OpenImages V4. (c) Ablative study on epistemic uncertainty to deep Q-evaluation.

Table 5: Ablation study on model design choices

| Model Design Choice | | | | | | MS COCO | | | | |
|:---:|:---:|:---:|:---:|:---:|:---:|:---:|:---:|:---:|:---:|:---:|
| IoU | Centerness | GIoU | GIoU+Uncertainty | DIoU | DIoU+Uncertainty | AP | $AP^S$ | $AP^M$ | $AP^L$ | $AP^{CH}$ |
| ✓ | ✗ | ✗ | ✗ | ✗ | ✗ | 44.6 | 28.7 | 45.9 | 52.1 | 25.8 |
| ✗ | ✓ | ✗ | ✗ | ✗ | ✗ | 42.4 | 26.2 | 45.8 | 51.6 | 23.7 |
| ✗ | ✗ | ✓ | ✗ | ✗ | ✗ | 44.3 | 28.5 | 45.6 | 51.5 | 25.4 |
| ✗ | ✗ | ✗ | ✓ | ✗ | ✗ | 46.7 | 30.2 | 47.5 | 53.4 | 28.1 |
| ✗ | ✗ | ✗ | ✗ | ✓ | ✗ | 45.4 | 29.5 | 46.8 | 52.4 | 26.7 |
| ✗ | ✗ | ✗ | ✗ | ✗ | ✓ | **47.6** | **31.0** | **48.5** | **54.3** | **28.9** |

**Results on RL baselines.** We also compare AIRS with existing RL based object detectors [47, 4, 3, 8, 43] in Table 4. It is worth to note that these works either focus on active object localization which can only detect limited objects or leverage CNN and recurrent networks to detect multiple objects step by step that lacks the flexibility to conduct dense detection from complex background. In [43], ReinforceNet leverages a hierarchical DRL framework for visual object tracking, which predicts target object's movement locations in the next frame given the last frame's state information. Since the primary goal is different from ours, their policy network is for mode switch among four modes (search, stop, update, re-initialization) given the last state. In contrast, our policy network gives directional movement actions given the current state (*i.e.,* patch location in the feature pyramid network) to support hierarchical search of objects. We ignore the specific $AP^S$, $AP^M$, $AP^L$ metrics and only report the mAP performance averaged over all categories, which is also the most commonly used metric on the Pascal VOC dataset. The result demonstrates the superiority of AIRS in dense scenarios with a large performance gap comparing to existing RL baselines.

### 4.4 Qualitative Analysis

Accurately selecting positive anchors via the generated RL masks has a strong impact on the final detection performance. For the more difficult images from the COCO and OpenImage V4 data sets, GFocal produces many false positive bounding boxes since it focuses on achieving a high recall on even small objects. In contrast, AIRS has precisely identified the true objects (high recall) while avoiding the unnecessary small bounding boxes (high precision). This phenomenon is supported by statistical counts in Figure 3 (a)-(b). This clearly justifies its effectiveness especially in those challenging images. Appendix D.7 shows more results in the even more challenging subsets.

### 4.5 Ablation Study

We demonstrate the effectiveness of each proposed component through ablation study on MS COCO dataset. Specifically, we analyze the impact of different model design choices, such as positive anchor criterion choices and the epistemic uncertainty design in evidential Q-learning. As shown in Table 5, among all positive anchor criterion choices, DIoU is the most effective positive anchor criterion as it considers both overlapping area ratio and the centerness relativity between the prediction and target. Furthermore, without epistemic uncertainty, there is a significant drop in detection performance. This justifies the importance of exploring the unknown patches by leveraging epistemic uncertainty in our proposed framework. In particular, for the challenging and small objects, it is more critical to conduct a deep exploration to identify various types of objects. The deeper reason justifying the importance of epistemic uncertainty can be explained using Figure 3c, where we use

one image's Q-learning curve in the search process as an example. The figure shows the Q-value (Q), evidential Q-value (Evi Q), epistemic uncertainty (EU) of the selected patch in each step, and the corresponding reward (Reward) as a supervision signal. We also include the Q-value and the corresponding reward in each step if we train the model without epistemic uncertainty, denoted as Q w/o EL (*i.e.,* evidential learning) and Reward w/o EL. Without epistemic uncertainty, at the early phase of training, the RL agent tends to select the patches with a high immediate reward and therefore abandons patches with a low estimated Q-value (with low immediate reward). As many patches that require a deep exploration to find objects may be missed, the agent only selects those regions with a high Q-value (shown in brown color) in the current step. Due to the top-down search strategy, the RL agent may never search the skipped region again, including all patches in the lower layers. This results in a significant reward drop (shown in the purple color) in the later steps along with an early termination (in step 40), resulting into a low cumulative reward in the long run. However, AIRS chooses regions based on the evidential Q-value (green curve) and therefore patches with even a low estimated Q-value (black curve) but high epistemic uncertainty (blue curve) may still have a chance to be selected. In this way, AIRS can explore patches with objects requiring deep exploration resulting into high cumulative rewards shown in red.

### 4.6   Discussion

**Performance of AIRS on large objects in MS COCO.** In this work, we aim to improve the detection performance by having a good balance between objects of different sizes and the $AP$ metric is designed to assess the overall effectiveness in terms of detecting objects in all granularities. Compared to competitive baselines, AIRS is superior on all datasets. We observe that by placing more focus on smaller and more difficult objects, AIRS achieves lower performance on $AP^L$ and $AP^M$ in MS COCO. However, this is an expected behavior as MS COCO has most of the objects being very large and therefore, the cost of missing smaller objects in the existing two-stage detectors seem to be low. As such, many two-stage detectors have superior performance (see Table 1). In contrast, as our technique leverages a one-stage detector to better cover dense objects, it is relatively less effective to detect very large objects (which is evidenced by the lower performance by all one-stage detectors in Table 1). It is worth mentioning that in other datasets, AIRS outperforms all baselines even on the large objects. In the case of Pascal VOC 2012, it is relatively easier and does not contain very large objects. As such, one-stage detectors perform comparable or even better than the two-stage detectors. As for Open Image V4, despite being challenging, it contains a good amount of training samples with larger objects, which provides enough supervision for models to detect these large objects. As such, all single-detectors including our technique perform comparable or even better compared to two-stage detectors.

**Comparison with two stage detectors like RPN.** There are key differences between two stage detectors and AIRS. The former usually relies on a Region Proposal Network (RPN), which is less effective to capture all targeted objects especially in a dense scenario. This is because, RPN selects anchors from the candidate anchors provided by the RPN based on the confidence score resulting into missing many true positive object anchors with a low confidence. In contrast, FPN in AIRS is based on multi-scale feature representations. Thus, the number of selected anchors in all layers is far more than the ones proposed by the RPN, which avoids missing important object anchors. To tackle the many false positive anchors in the FPN based approaches, we propose a novel hierarchical search mechanism coupled with an effective exploration-exploitation strategy leveraging evidential Q-learning. As a result, AIRS effectively removes the false positive bounding boxes without removing the less confident true positive objects. This phenomenon is also demonstrated in Table 1, where two-stage detectors result in a lower performance compared to AIRS in dense object detection.

## 5   Conclusion

We propose a novel Adaptive Important Region Selection (referred to as AIRS) framework guided by evidential Q-learning built upon a uniquely designed reward function. AIRS encourages object search in a hierarchical, top-down fashion, where the RL agent moves down to a fine-grained level only when it is likely to contain an object of interest. In addition, to facilitate detection of unknown patches, evidential Q-learning leverages the epistemic uncertainty to guide the exploration process. Our proposed technique dynamically balances exploration-exploitation where in the early phase the priority is given to the highly uncertain patches and in the latter phase priority is dynamically shifted to the potentially positive patches. Both theoretical analysis and empirical results on challenging object detection datasets demonstrate the effectiveness of our proposed framework.

## Acknowledgments

This research was supported in part by an NSF IIS award IIS-1814450. The views and conclusions contained in this paper are those of the authors and should not be interpreted as representing any funding agency. We would like to thank the anonymous reviewers for their constructive comments.

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

# Supplementary Material

# Appendix

## Table of Contents

## Organization of the Appendix:

- In Appendix A, we summarize major notations used throughout the paper.
- In Appendix B, we provide detailed proof of the main theoretical result.
- In Appendix C, we provide more detailed information about our `AIRS` framework including reward design and action interaction.
- In Appendix D, we provide details related to experimentation along with additional results.
- In Appendix E, we discuss the limitation of our work and its broader impact.
- In Appendix F, we provide the link to the source code.

## A    Summary of Notations

The major notations and their descriptions are summarized in Table 6.

Table 6: Symbols with Descriptions

| Notation | Description |
|---|---|
| $p_{i,t}^l$ | $i$th selected region in $l$th layer at time step $t$ |
| $P_{s_t,a_t}^l$ | The penalty term of the next selected patch on layer $l$ in time step $t$ |
| $(x_{i,t}^l, y_{i,t}^l)$ | Center of region $p_{i,t}^l$ |
| $i, L$ | Index of sub-patch and top layer index in FPN |
| $N_{epoch}, n_{epoch}$ | The number of total epoch and current epoch |
| $T$ | The number of maximum time step |
| $D$ | Dimensionality of action space |
| $\mathbf{s}_t$ | State space representation in time step $t$ |
| $\mathbf{e}_t$ | Feature embedding representation in time step $t$ |
| $\mathbf{a}_{d,t}$ | Action of movement signal $d$ at time step $t$ |
| $r(\mathbf{s}_t, a_t)$ | Reward associated with the state $\mathbf{s}_t$ generated by action $\mathbf{a}_{d,t}$ |
| $\gamma$ | Discounting factor used in the TD computation |
| $\mathcal{P}$ | MDP transition matrix |
| $q_{d,t}$ | Q-value estimates over the action space $\mathbf{a}_t$ |
| $\widehat{q_{d,t}}$ | Temporal different target associated with the action d in time step t |
| $q_{d,t}^e$ | Evidential Q-value estimate associated with the action d in time step t |
| $\widetilde{\mathbf{q}_{d,t}^e}$ | Masked evidential Q-value estimate associated with the action d in time step t |
| $\mathbf{m}_{l,t}^d$ | Mask value (binary) associated with the $d^{th}$ action in $l^{th}$ layer in $t^{th}$ time step |
| $\boldsymbol{\theta}_f$ | Parameters associated with the Feature Extractor |
| $\boldsymbol{\theta}_r$ | Parameters associated with the state encoder (RNN) |
| $\boldsymbol{\theta}_e$ | Parameters associated with Evidential Q-network |
| $\Theta$ | All network parameters in our framework |
| $\mu_{d,t}$ | Mean of the Gaussian distribution |
| $\sigma_{d,t}^2$ | Variance of the Gaussian distribution |
| $\nu_{d,t}$ | Hyper-parameter of Evidence network as an output from Evidential Q-network |
| $\beta_{d,t}$ | Hyper-parameter of Evidence network as an output from Evidential Q-network |
| $\gamma_{d,t}$ | Hyper-parameter of Evidence network as an output from Evidential Q-network |
| $\alpha_{d,t}$ | Hyper-parameter of Evidence network as an output from Evidential Q-network |
| Inv-Gamma$(\cdot)$ | Inverse Gamma function |
| $\lambda$ | Hyper-parameter balancing the exploration and exploitation |

## B    Proof of Theoretical Results

We aim to obtain an upper bound for $\|Q^{\pi_K} - Q^*\|_1$. We start by providing formal definitions of several key components used in our proof, including sparse ReLU network, Hölder smooth functions,

their compositions, and functional classes. We then provide two assumptions related to Bellman optimality smoothness and concentration coefficient bound. Finally, we provide the proof for Theorem 1. It should be noted that our proof is developed based upon some key results in [35, 2, 13].

## B.1 Definitions and Assumptions

**Definition B.1** (Sparse ReLU Network). Let $L$ be the number of hidden layers, $V$ be the upper bound for the neural network outputs, $d_l$ be the number of nodes in the $l^{th}$ layer, $f_l$ be the output of the $l$-th layer, and $w_{max}$ be the total number of non-zero weights. The sparse ReLU networks can be formally defined as

$$F\left(L, \{d_l\}_{l=0}^{L+1}, w_{max}, V\right) = \left\{f : \max_{l \in [L+1]} \|\widetilde{\mathbf{W}}_l\|_\infty \le 1, \sum_{l=1}^{L+1} \|\widetilde{\mathbf{W}}_l\|_0 \le w_{max}, \max_{l \in [d_{L+1}]} \|f_l\|_\infty \le V \right\} \tag{10}$$

In our case, since Q-values are always bounded and therefore, we can replace $V$ by $V_{max} = \frac{R_{max}}{1-\gamma}$. To simplify the notations, we can simply omit the $V$ term and use notation $F(L, \{d_l\}_{l=0}^{L+1}, w_{max})$. In the above equation, $\widetilde{\mathbf{W}}_l$ denotes the weights of the $l^{th}$ layer of the neural network, *i.e.,*, $\widetilde{\mathbf{W}}_l = (\mathbf{W}_l, b_l)$ and the final output $f$ of the network is given by

$$f(\mathbf{x}) = \mathbf{W}_{L+1}\sigma(\mathbf{W}_L\sigma(\mathbf{W}_{L-1}.....\sigma(\mathbf{W}_2\sigma(\mathbf{W}_1\mathbf{x} + b_1) + b_2)...b_{L-1}) + b_L) \tag{11}$$

where $\sigma$ is the ReLU function. In our case, we can have a ReLU activation in each output layer except for the last layer to make the network sparse, which is required to prove the theorem. In the later part of our proof given by Equation 31, we show that the number of samples and network sparsity are directly related (more sparse, less samples) and therefore, to get the given convergence result, having a sparse network will reduce the number of samples.

**Definition B.2** (Hölder Function). Let $\mathcal{D}$ be a compact subset of $\mathcal{R}^r$. Then, the set of Hölder functions on $\mathcal{D}$ is defined as

$$\mathcal{C}_r(\mathcal{D}, \nu, H) = \left\{f : \mathcal{D} \to \mathcal{R} : \sum_{\boldsymbol{\alpha}:|\boldsymbol{\alpha}|<\nu} \|\delta^{\boldsymbol{\alpha}} f\|_\infty + \sum_{\boldsymbol{\alpha}:\|\boldsymbol{\alpha}\|_1=\nu_0} \sup_{\mathbf{x},\mathbf{y}\in\mathcal{D}, \mathbf{x}\neq\mathbf{y}} \frac{|\delta^{\boldsymbol{\alpha}} f(\mathbf{x}) - \delta^{\boldsymbol{\alpha}} f(\mathbf{y})|}{\|\mathbf{x} - \mathbf{y}\|_\infty^{\nu-\nu_0}}\right\} \tag{12}$$

where $\nu > 0, H > 0$ are function parameters, $\nu_0$ is the largest integer no greater than $\nu$, $\boldsymbol{\alpha} = (\alpha_1, ..., \alpha_r)^\top$, $\delta^{\boldsymbol{\alpha}} = (\delta^{\alpha_1}, ...\delta^{\delta_r})$, and $r$ is the dimensionality of the state space $\mathcal{S}$.

**Definition B.3** (Compositions of Hölder Function). Let $q$ and $\{p_j\}_{j\in[q]}$ be integers and $\mathbf{g}_j$ be a function with $\mathbf{g}_{jk}$ being a Hölder smooth function that depends on at most $\eta_j$ components of its input, *i.e.,*, $\mathbf{g}_{jk} \in \mathcal{C}_{\eta_j}(D_j, \nu_j, H_j)$. With $G(\{p_j, \eta_j, \nu_j, H_j\}_{j\in[q]})$ being the family of functions that can be expressed as compositions of $\{g_j\}_{j\in[q]}$ then for any $f \in G(\{p_j, \eta_j, \nu_j, H_j\}_{j\in[q]})$, we can write the following

$$f = \mathbf{g}_q \cdot \mathbf{g}_{q-1} \cdot .... \cdot \mathbf{g}_2 \cdot \mathbf{g}_1 \tag{13}$$

**Definition B.4** (Functional Classes). Let $F(L, \{d_l\}_{l=1}^{L+1}, w_{max})$ be the family of sparse ReLU networks defined on the state space $\mathcal{S}$ with $d_0 = r$ and $d_{L+1} = 1$, then $\mathcal{F}_0$ can be defined as

$$\mathcal{F}_0 = \{f : \mathcal{S} \times \mathcal{A} \to \mathcal{R} : f(., \mathbf{a}) \in F(L, \{d_l\}_{l=0}^{L+1}, w_{max}); \forall \mathbf{a} \in \mathcal{A}\} \tag{14}$$

In addition, let $G(\{p_j, \eta_j, \nu_j, H_j\}_{j\in[q]})$ be a set of compositions of the Hölder smooth functions defined on $\mathcal{S} \in \mathcal{R}^r$. Similar to $\mathcal{F}_0$, we define functional class $\mathcal{G}_0$ as

$$\mathcal{G}_0 = \{f : \mathcal{S} \times \mathcal{A} \to \mathcal{R}; f(., \mathbf{a}) \in G(\{p_j, \eta_j, \nu_j, H_j\}_{j\in[q]}); \forall \mathbf{a} \in \mathcal{A}\} \tag{15}$$

Based on these definitions, we introduce two assumptions about the Hölder smoothness on the Bellman optimality and concentration coefficient bound, which are given below.

**Assumption B.5** (Bellman Optimality Smoothness). For any sparse ReLU network function $f \in F$ with $F$ being the family of functions, $(Tf)(\mathbf{s}, \mathbf{a})$ with $T$ being the Bellman optimality operator can be written as the composition of the Hölder smooth functions.

$$(Tf)(\mathbf{s}, \mathbf{a}) = \mathbf{g}_q \cdot \mathbf{g}_{q-1} .... \cdot \mathbf{g}_1 \cdot \mathbf{g}_0 \tag{16}$$

In the above equation, $\mathbf{g}_j$ has $p_{j+1}$ components, where each component $g_{jk}$ is a Hölder smooth function as defined above.

**Remark about Assumption 1.** It should be noted that Bellman Optimality Smoothness assumption holds whenever the reward function is smooth [13]. As our reward function is based on existing metrics (e.g., IoU, GIoU), we can easily convert to a smooth variant [34]. This will make the Assumption 1 true in our proposed framework.

**Assumption B.6** (Concentration Coefficient Bound). Let $u_1, u_2 \in \mathcal{P}(\mathcal{S}, \mathcal{A})$ be two probability measures absolutely continuous with respect to the Lebesgue measure on $\mathcal{S} \times \mathcal{A}$. Also consider $\{\pi_t\}_{t \geq 1}$ to be a sequence of policies and for any integer $m$, let us denote the distribution of $\{\mathcal{S}_t, \mathcal{A}_t\}_{t=0}^m$ by $P^{\pi_m} P^{\pi_{m-1}} ..... P^{\pi_1} u_1$. Then, the $m^{th}$ concentration coefficient is given as

$$\kappa(m; u_1, u_2) = \sup_{\pi_1, ..., \pi_m} \left[ \mathbb{E}_{u_2} \left| \frac{d(P^{\pi_m} P^{\pi_{m-1}} .... P^{\pi_1} u_1)}{du_2} \right| \right] \tag{17}$$

We assume that there exists a constant $\phi_{e,f} \leq \infty$ that bounds the concentration coefficient, given as

$$(1 - \gamma^2) \sum_{m \geq 1} \gamma^{m-1} m \kappa(m; e, f) \leq \phi_{e,f} \tag{18}$$

where $e$ is the fixed distribution of $\mathcal{S} \times \mathcal{A}$ and $f$ is the sampling distribution.

**Remark about Assumption 2.** The Concentration Coefficient Bound assumption is commonly used in a large class of MDP systems [31]. It requires sampling distribution $f$ to have sufficient coverage over $\mathcal{S} \times \mathcal{A}$. In our context, because of the novel exploration strategy coupled with hierarchical searching strategy, our sampling distribution will likely to have a sufficient coverage on $\mathcal{S} \times \mathcal{A}$, implying that this assumption holds.

## B.2 Proof of Theorem 1

Based on the above assumptions, we proceed to prove Theorem 1, which is restated as:

$$\|Q^* - Q^{\pi_K}\|_1 \leq C\phi_{e,f} \frac{\gamma}{1 - \gamma^2} |\mathcal{A}| \tau(\eta, \nu) + \frac{4\gamma^{K+1}}{1 - \gamma^2} R_{max} \tag{19}$$

*Proof.* In the proposed AIRS algorithm, $\pi_k$ is the policy with respect to $Q^{\pi_k}$ and let $Q^{\pi_K}$ be the action-value function associated with $\pi_K$. Since, $\{Q^{\pi_k}\}_{k \in [K]}$ is constructed by an iterative algorithm, it is helpful to relate $\|Q^* - Q^{\pi_K}\|_1$ to the errors that occur in the previous steps in AIRS, *i.e.,* $\{Q^{\pi_k} - TQ^{\pi_{k-1}}\}_{k \in [K]}$. Therefore, in the first step, we provide the upper bound for $\|Q^* - Q^{\pi_K}\|_1$ as a function of error introduced in each step, which is given by the following theorem.

**Theorem B.7.** *The relationship between* $\|Q^* - Q^{\pi_K}\|_1$ *and* $\|Q^{\pi_k} - TQ^{\pi_{k-1}}\|_f$ *with* $f$ *being the sampling distribution is*

$$\|Q^* - Q^{\pi_K}\|_1 \leq \frac{2\phi_{e,f}\gamma}{1 - \gamma^2} \max_{k \in [K]} \|Q^{\pi_k} - TQ^{\pi_{k-1}}\|_f + \frac{4\gamma^{K+1}}{(1 - \gamma^2)} R_{max} \tag{20}$$

Please refer to Section C.1 of [13] for a detailed proof of the above theorem. It should be noted that the first term (specifically $\max_{k \in [K]} \|Q^{\pi_k} - Q^{\pi_{k-1}}\|_f$) is a statistical error and the second term is an algorithmic error. In the later part of our proof, we show that the statistical error diminishes as the sample size $n$ in each iteration grows whereas the algorithmic error decays to zero geometrically with $K$. In the above equation, to get the upper bound, we need to bound $\|Q^{\pi_k} - TQ^{\pi_{k-1}}\|_f$. In order to do this, we use the nonparametric regression. Specifically, we provide the following theorem to bound that error.

**Theorem B.8.** *Under the assumption of the Bellman optimality smoothness in Assumption 1, for any* $k \in [K]$, *we have*

$$\|Q^{\pi_{k+1}} - TQ^{\pi_k}\|_f^2 \leq 4[dist_\infty(\mathcal{F}_0, \mathcal{G}_0)]^2 + \frac{CR_{max}^2}{n(1-\gamma)^2} \log N_\delta + \frac{CR_{max}}{1-\gamma} \delta \tag{21}$$

*For any* $\delta > 0$ *and* $C > 0$ *being a constant.*

Please refer to the Section C.2 of [13] for a detailed proof for the above theorem. There are two terms that play a major role in the bound. The first term indicates the bias, which is given as

$$dist_\infty(\mathcal{F}_0, \mathcal{G}_0) = \sup_{f' \in \mathcal{G}_0, f \in \mathcal{F}_0} \|f - f'\|_\infty \tag{22}$$

This term reflects the $l_\infty$ error of estimating the function in $\mathcal{G}_0$ using the sparse ReLU defined in $\mathcal{F}_0$. It also indicates the bias in estimating the functions in $\mathcal{G}_0$. The second term $N_\delta$ indicates the minimum cardinality of the balls required to cover function $\mathcal{F}_0$ with respect to the $l_\infty$-norm. This term indicates the variance associated with estimating the action-value function using a sparse ReLU network. Substituting $\delta = \frac{1}{n}$, we can rewrite the above equation as

$$\|Q^{\pi_{k+1}} - TQ^{\pi_k}\|_f^2 \leq 4 dist_\infty^2(\mathcal{F}_0, \mathcal{G}_0) + \frac{CR_{max}^2}{n(1-\gamma^2)} \log N_\delta + \frac{CR_{max}}{(1-\gamma)n}$$

In the Above Equation, in the right hand side, the last term is constant for a given n. Then there exists absolute constant $C' > 0$ such that following Equation holds

$$\frac{CR_{max}^2}{n(1-\gamma^2)} \log N_\delta + \frac{CR_{max}}{(1-\gamma)n} \leq C' \frac{R_{max}^2}{n(1-\gamma)^2} \log N_\delta$$

Using this inequality, we have the following

$$\|Q^{\pi_{k+1}} - TQ^{\pi_k}\|_f^2 \leq 4 dist_\infty^2(\mathcal{F}_0, \mathcal{G}_0) + \frac{C'R_{max}^2}{n(1-\gamma)^2} \log N_\delta, \ C' > 0 \tag{23}$$

Now, if we establish a bound for $dist_\infty(\mathcal{F}_0, \mathcal{G}_0)$ and $\log N_\delta$, we will be able to get the bound for the $\|Q^{\pi_k} - TQ^{\pi_{k-1}}\|_f$ as well. So let us find out the bound for each term.

**(1) Bound for $dist_\infty(\mathcal{F}_0, \mathcal{G}_0)$:** To get the bound for $dist_\infty(\mathcal{F}_0, \mathcal{G}_0)$, we first show that ReLU network $f(., \mathbf{a})$ can be reformulated as a composition of Hölder functions defined on the hypercube. Next, we show that using Lemma 6.3 in [35], we can construct the ReLU network to approximate the hypercube components yielding a function close to $f(., \mathbf{a})$ in the $l_\infty$ norm. Thus, $f(., \mathbf{a})$ can be reformulated as a compositions of Hölder functions defined on the hypercube. Considering, $\mathbf{h}_1 = \frac{\mathbf{g}_1}{2H_1}$, $\mathbf{h}_q(u) = \mathbf{g}_q(2H_{q-1}u - H_{q-1})$ and $\mathbf{h}_j(u) = \frac{\mathbf{g}_j(2H_{j-1}u - H_{j-1})}{2H_j} + \frac{1}{2}; \forall j \in [2, q-1]$, we can write

$$f(., \mathbf{a}) = \mathbf{g}_q \cdot .... \cdot \mathbf{g}_1 = \mathbf{h}_q \cdot .... \cdot \mathbf{h}_1 \tag{24}$$

where we can write $h_{jk} \in \mathcal{C}_{\eta_j}([0,1]^{\eta_j}, W)$ with $W > 0$ and given as

$$W = \max \left\{ \max_{1 \leq j \leq q-1} (2H_j - 1)^{\nu_j}, H_q(2H_q - 1)^{\nu_q} \right\} \tag{25}$$

Now we can use Lemma 6.3 from [35] to construct a ReLU network to approximate $h_{jk}$, which can be combined with Equation 24 to show that a ReLU network can be used to approximate $f(., \mathbf{a})$ in the $l_\infty$ norm. According to Lemma 6.3 from [35], there exists a ReLU network $\tilde{h}_{jk}$ which is Hölder smooth such that $\|\tilde{h}_{jk} - h_{jk}\|_\infty \leq N^{-\frac{\nu_j}{\eta_j}}$. When $N$ is large, it can be written as

$$N = \left\lceil \max_{1 \leq j \leq q} C n^{\frac{\eta_j}{2(\nu_j^* + \eta_j)}} \right\rceil \tag{26}$$

where $\nu_j^* = \nu_j \prod_{l=j+1} \min(\nu_l, 1)$ with $\nu_j^* = 1$. For the large $N$, we can approximate $f(., \mathbf{a})$ by $\tilde{f}$ belonging to the ReLU class $F(L^*, \{d_j^*\}_{j=1}^{L^*+1}, w_{max}^*)$. Specifically, we have the following bound for the approximation.

$$\|f(., \mathbf{a}) - \tilde{f}\|_\infty \leq \sum_{j=1}^{q} \|\tilde{h}_j - h_j\|_\infty^{\lambda_j} \tag{27}$$

where, $\lambda_j = \prod_{l=j+1}^{q} (\eta_l \wedge 1) \forall j \in [q-1]$ with $\lambda_q = 1$. Now using above inequality along with definition of $N$ and $\|\tilde{h}_{jk} - h_{jk}\|_\infty \leq N^{-\frac{\nu_j}{\eta_j}}$, we get the following

$$[dist_\infty(\mathcal{F}_0, \mathcal{G}_0)]^2 \leq n^{\alpha^* - 1} \tag{28}$$

where $\alpha^* = \max_{j \in [q]} \frac{\eta_j}{2\nu_j^* + \eta_j}$ and $\nu_j^* = \nu_j \prod_{l=j+1} \min(\nu_l, 1)$.

**(2) Bound for** $\log N_\delta$**:** Using classical results on the covering number of neural networks in [2], we have the following

$$\log N_\delta \leq |\mathcal{A}| w_{max}^* L^* \max_{j \in [L^*]} \log(d_j^*) \tag{29}$$

Let us consider that there exists $\zeta$ with $\zeta^* = 1 + 2\zeta$ such that

$$\max \left\{ \sum_{j=1}^{q} (\eta_j + \nu_j + 1)^{3+\eta_j}, \sum_{j \in [q]} \log(\eta_j + \nu_j), \max_{j \in [q]} p_j \right\} \leq (\log n)^\zeta \tag{30}$$

We further assume that

$$L^* \leq (\log n)^{\zeta^*}, r \leq \min_{j \in [L^*]} d_j^* \leq \max_{j \in [L^*]} d_j^* \leq n^{\zeta^*}, w_{max}^* \approx n^{\alpha^*} (\log n)^{\zeta^*} \tag{31}$$

Then we can rewrite Equation 29 as

$$\log N_\delta \leq |\mathcal{A}| n^{\alpha^*} (\log n)^{1+2\zeta^*} \tag{32}$$

Now substituting Eq. 32 and 28 into Eq. 23 and replacing it into Eq. 20, we obtain the following

$$\|Q^* - Q^{\pi_K}\|_1 \leq \frac{C\phi_{e,f}\gamma}{1-\gamma^2} |\mathcal{A}| (\log n)^{1+2\zeta^*} n^{\frac{\alpha^*-1}{2}} + \frac{4\gamma^{K+1}}{(1-\gamma^2)} R_{max} = \frac{C\phi_{e,f}\gamma}{1-\gamma^2} |\mathcal{A}| \tau(\nu, \eta) + \frac{4\gamma^{K+1}}{(1-\gamma^2)} R_{max} \tag{33}$$

This completes the proof of our theorem. □

## C   Additional Details of AIRS

In this section, first we introduce the reward design in detail. Then we explain the overall detailed training and test process through the complementary diagram and pseudo code. Next, we explain different actions defined and their interactions with the RL environment (*i.e.,* FPN tree-structure) under `AIRS` framework with some illustrative examples.

### C.1   Reward Design

For the reward design, first we need to get positive anchor by using IoU, GIoU, DIoU, etc as selection criterion, along with the corresponding thresholds following RetinaNet [26], we did an ablation study for these choices in Table 5. Then we calculate the number of positive anchors for each patch and rank those patches in the same layer to form $L$ ranking lists. If the action directed next patch in any list resides on 0-25th percentile, the quality score $g$ is up-scaled to 0.25, or if it resides on 75th percentile to 100th percentile, the quality score is up-scaled to 1, etc. In this way, the quality score and the penalty term have an alignment and perform exploitation-exploration balance in the reward design besides evidential Q-learning.

### C.2   Training/Test process

Fig 5 shows the detailed workflow of `AIRS`. It also captures important steps of the training process along with the major symbols and parameters used in the actual implementation. For RL training, we leverage the FPN from the backbone model (*e.g.,* pre-trained GFocal) to set up the RL environment. Training starts from the initial patch $p_{i,t}^l$ on the virtual top-most layer $L$, where state embedding representation $s_t$ is generated after passing through the feature extractor $f(\cdot; \boldsymbol{\theta}_f)$ and the RNN network. Next, the state is passed into the evidential Q-network to get hyper-parameters governing the evidential distribution, where Q-value is sampled. After getting the Q-value $q_{d,t}$ and epistemic uncertainty $u_{d,t}$ generated from the same distribution, we evaluate the evidential Q-value $q_{d,t}^e$. Then, by imposing environment constraints through masking distribution, the RL agent selects the next action $a_t$ based on maximum masked evidential Q-value $\widetilde{q_{d,t}^e}$. Finally, the RL agent moves to next step's patch $p_{i^*,t+1}^{l^*}$, calculates reward $r_t$, and collects training tuples $(s_t, a_t, r_t, s_{t+1})$ along this process.

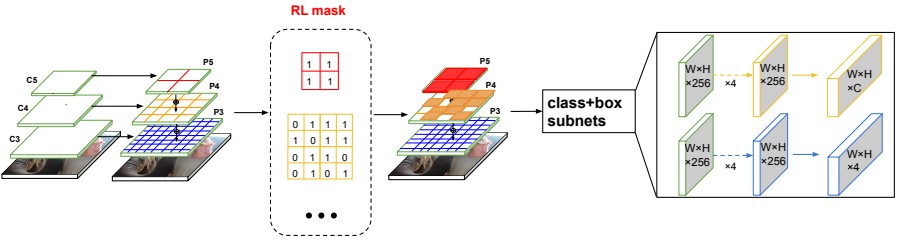

Figure 4: RL-augmented detection process

It keeps searching in a given image until the upward movement action is selected in the highest $L$ layer or maximum time step $T$ is reached. After every $K$ such iterations, `AIRS` samples one batch of training tuples from the replay buffer $\mathcal{D}$ and conducts off-policy Q-learning to train all network parameters $\Theta$. The overall training process is shown in Algorithm 1, where we use $\Theta = \{\boldsymbol{\theta}_e, \boldsymbol{\theta}_f, \boldsymbol{\theta}_r\}$ to denote all network parameters. In test phase, as shown in Fig. 4, those aforementioned binary RL masks will be used to mask out unnecessary patches across different levels in the FPN structure so that those low-value false positive anchors (candidate regions) will be filtered out and never passed to the head blocks for the bounding box prediction.

---

**Algorithm 1** `AIRS` Training

---

**Require:** Hyperparameters: $N_{epoch}, T, K, \eta, \gamma, \lambda, lr$
1: Initialize network parameters ($\Theta$), epoch $n_{epoch} = 0$, stack $S = []$, current selected patch $p_{i,t}^l = p_{*,t}^L$, action $\mathbf{a}_t = \mathbf{0}$, count of training tuple $k = 0$
2: **repeat**
3:     **repeat**
4:         time step $t = 0$,
5:         **repeat**
6:             Generate embedding $\mathbf{e}_t \leftarrow \mathrm{f}(p_{i,t}^l; \boldsymbol{\theta}_f)$
7:             Compute state $\mathbf{s}_t$ per $\mathbf{s}_t = \mathrm{RNN}(\mathbf{e}_t, \mathbf{s}_{t-1}; \boldsymbol{\theta}_r)$
8:             Compute evidential Q-value estimate $\mathbf{q}_{d,t}^e$ per Eq. (4)
9:             Compute masked evidential Q-value per Eq. (5)
10:            Update action $\mathbf{a}_t$ per Eq. (6)
11:            Select next patch $p_{i^*,t+1}^{l^*}$ from RL environment given current patch and new action $\mathbf{a}_t$
12:            Add last patch $p_{i,t}^l$ to the stack $S$ recording visited patches
13:            Compute RL reward $r_t$ based on Eq. (8)
14:            Collect the training tuple $s_t, a_t, r_t, s_{t+1}$ into replay buffer $\mathcal{D}$
15:            $t = t + 1, k = k + 1$
16:         **until** $t > T$ or ($p_{i,t}^l == p_{*,t}^L$ and $a_{D-1,t} = 1$)
17:         **if** $k\%K = 0$ **then**
18:            Compute total loss $\mathcal{L}_\Theta$ using Eq. (7)
19:            Update $\Theta \leftarrow \Theta - lr \times \frac{\delta \mathcal{L}_\Theta}{\delta \Theta}$
20:         **end if**
21:     **until** One Epoch Ended
22:     $n_{epoch} = n_{epoch} + 1$
23: **until** $n_{epoch} > N_{epoch}$

---

### C.3   Action Interaction Details

Given the current selected patch $p_{i,t}^l$ and an action space with size $D$, the first $D-1$ actions $a_{d,t}, d \in [0, \cdots, D-2]$ denote the downward movements directing into one of the sub-patches $p_{d,t+1}^{l-1}$ when $a_{d,t} = 1$; the last action $a_{D-1,t}$ denotes an upward movement into the mother patch from the immediate upper layer. An upward movement in the initial patch $p_{*,t}^L$ of highest level $L$ indicates

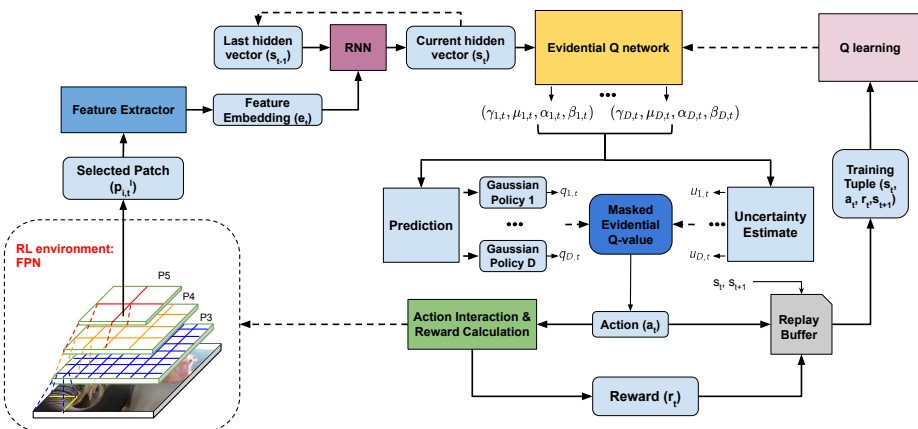

Figure 5: Detailed Workflow of `AIRS`

the termination of RL searching for one image input. The Action Interaction module determines the constraint mask by considering the constraints from RL environment, *i.e.,* we mask the evidential Q-value of any downward movement, which directs to an already visited sub-patch or void space (*e.g.,* downward movements on bottom layer 0) to zero. Specifically, we generate a mask $\mathbf{m}_{l,t}$ of length $D$, where $\mathbf{m}_{l,t}^d$ is the individual unit contained in the mask. For downward movements $\mathbf{m}_{0,t}^i, i \in [0,1,2,3]$ on the bottom layer or directing to visited patches before, we set the mask value to zero to avoid such illegal actions per above instructions. Then, the masked evidential Q-value is given by performing element-wise multiplication between $\mathbf{q}_{d,t}^e$ and $\mathbf{m}_{l,t}^d$.

**Examples Demonstrating Action Interaction Process:**  We follow a depth-first-search (DFS) rule to conduct the hierarchical search in the FPN tree-structure and further include several case studies as examples to illustrate the interaction protocol under different selected actions. Let $(x_{i,t}^l, y_{i,t}^l)$ be the center (also an alias) of the $i^{th}$ region in the $l^{th}$ layer at time step $t$. In the beginning (*i.e.,* step $t = 0$), we pass all the regions present in the $L$-layer to obtain the masked evidential Q-value $\widetilde{\mathbf{q}_{d,t}^e}$ for each action $a_{d,t}$ using Eq. (5) and then select the next action corresponding to the maximum masked evidential Q-value estimation. Depending on the action selected, we define the -following three cases that describe the corresponding behavior of the agent to form the interaction protocol.

**Case 1: Downward movement.**  It happens when $\sum_{d=0}^{D-2} a_{d,t} = 1$. The RL agent will go down to one of the lower-level regions from the current region. For example, in time step $t = 0$, assuming the uppermost virtual layer as the current region (*i.e.,* root of the hierarchy), the above condition directs the agent to go into one of the sub-patches from the P5 layer. Out of $D - 1$ regions, which one to go is determined by Equation 6. For example in time step $t = 0$ of Figure 6, we have $\mathbf{a}_{t=0} = (1,0,0,0,0)^\top$, which means the agent will move to the top-left region $(x_{0,t=1}^{L-1}, y_{0,t=1}^{L-1})$ in P5. Next, in time step $t = 1$, we pass the newly selected region as an input to the network and again obtain the action value $\mathbf{a}_{t=1} = (0,1,0,0,0)^\top$, then the RL-agent will visit the top-right child region $(x_{1,t=2}^{L-2}, y_{1,t=2}^{L-2})$ in P4 associated with the top-left region $(x_{0,t=1}^{L-1}, y_{0,t=1}^{L-1})$ from P5. We then pass the top-right region from P4 as an input in time step $t = 2$. This process continues until the agent needs to move upwards or stop the search.

**Case 2: Upward movement.**  It happens when $a_{D-1,t} = 1$. The RL agent will stop going further down in the hierarchy as the evidential Q-value indicates that no valuable information is available in the finer level of granularity to cover the searching cost. Thus, the agent will go back to the parent region which allows the agent to search other sibling patches. For example, in time step $t = 2$ of Figure 6, we have $\mathbf{a}_{t=2} = (0,0,0,0,1)^\top$ and the agent goes back to the parent region, which is top-left region in P5. It should be noted that we have used the mask generator to avoid the action choices directing to already visited and out-of-boundary regions. Next in time step $t = 3$, the top-right region in P4 is already visited and therefore, masked evidential Q-value estimate $\widetilde{\mathbf{q}}_{d=1,t=3}^e$ is

made zero. As shown in time step $t = 3$ of Figure 7, while this region has the highest evidential Q-value among P4 patches, we select the second best region, which is the top-left one from P4. In time step $t = 4$ of Figure 7, the input region becomes the top-left region from P4 associated with the top-left region from P5 and the search process continues.

**Case 3: Search termination.** It happens when upward movement $a_{D-1,t} = 1$ happens in the uppermost virtual layer which is the root node of the FPN tree-structure or when maximum time step $T = 60$ has been reached. In such cases, the RL agent is given the stop signal to terminate the entire search process, which implies that a sufficient number of high quality regions have been detected.

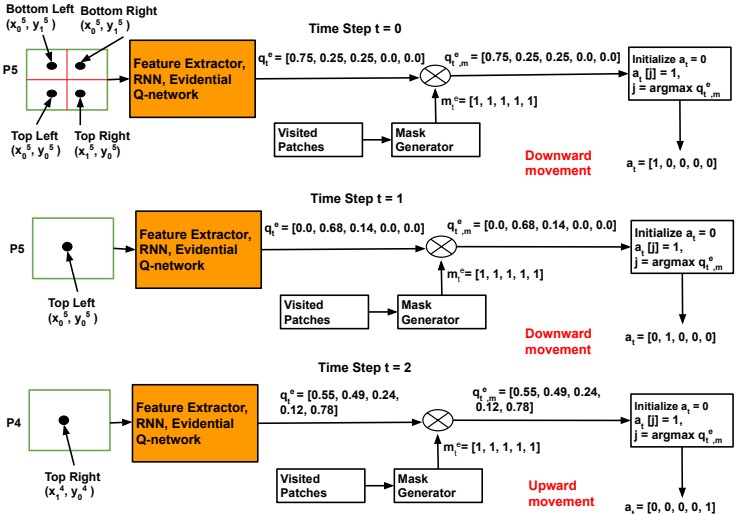

Figure 6: Action interaction process: time steps $t = 0, 1, 2$

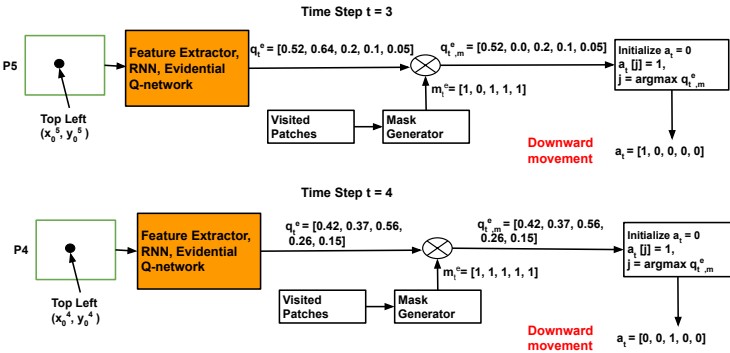

Figure 7: Action interaction process: time steps $t = 3, 4$

### C.4 Clarification on RL masks.

**Clarification on use of masks to generate final bounding boxes.** We run the trained RL agent on the test image's FPN to generate RL masks. Based on the masked evidential Q-value estimate, the agent selects the next action, which would be either a downward or upward movement. Then, the agent moves to the next patch and continues the process until receiving an upward movement in layer $L$ or reaches the maximum time step i.e. $T$. After the hierarchical searching process, we

can have a binary RL mask by recording which patches are visited by RL agent through it actions (denoted as 1 in RL mask), and which are not visited by the RL agent (denoted as 0 in RL mask). Given the RL mask, which is a three-level binary mask covering the feature pyramid network (FPN), each pixel in the FPN will be assigned a confidence score in the quality evaluation branch to decide if it is a positive anchor or not by (comparing with a threshold). Those pixels covered by the zero RL masks will have their confidence score reset to 0, and other pixels will maintain the same confidence score. In that way, RL masks serve as an additional filter to further eliminate the "false positive" bounding boxes.

**How the binary RL mask helps reduce unnecessary bounding boxes in inference step.** We are handling two different types of false positive bounding boxes. The first category involves bounding boxes that capture only background with no targeted object. To remove those patches, our novel exploration-exploitation strategy plays a major role. Specifically, during the adaptive hierarchical search in the top-down fashion, exploration of the higher layer quickly discovers that there is no object in the lower-level granularity. Specifically, both Q-value as well as epistemic uncertainty (see Eq. 4) remains low, leading to removing bounding boxes on backgrounds. The second category involves bounding boxes that cover only a part of a given object and are embedded in the larger bounding box that covers the whole object. As our approach works in the top-down fashion, once the RL agent explores the bigger bounding box covering the full object, the model assigns a very low epistemic uncertainty for partially covering bounding boxes. As such, the model avoids going downward in a lower level granularity in the action space resulting in removing unnecessary partially covered bounding boxes.

# D    Additional Experiments

In this section, we first present the additional comparison results with YOLO series. After that, we show additional ablation study results that investigate the impact of the underlying base detectors and balancing hyper-parameters. We also test the transferred performance of the RL masks trained from the proposed `AIRS` and apply them to other base detectors. Finally, we show some additional quantitative and qualitative results on the challenging datasets (aerial park lot [16]) or subsets chosen from all three data sets, containing difficult images with a large amount of small objects.

## D.1    Comparison with the Latest YOLO Series

In this set of experiments, we include the latest YOLO series for comparison. We run the experiments for three times with different random seeds to verify the performance and provide the strongest YOLO-V7 comparison results with statistical significance in Table 7. Note that we use the same hyper-parameters reported in the original paper, including the image augmentation, learning rate, momentum decay, etc. For a fair comparison, we train all the baselines until convergence and test the models on the same test splits from different datasets, and we align all the weight initialization to be Xavier initialization. As can be seen, comparing to these SOTA models, for the overall AP, `AIRS` is better than any YOLO series below medium parameter size scale and only slightly lower than yolo V6-L, yolo V7 on MS COCO (but significantly better than them on Open Image V4). It shows a clear advantage in images with small objects and with difficult dense scenarios, which is achieved by a good balance of recall and precision thanks to the FPN with RL selected mask augmentation. The good AP performance from yolo V6, V7 on MS COCO is likely due to the special architectural design optimization targeting this dataset, where medium and large objects form the majority of image labels. These include extended efficient layer aggregation networks, model scaling for concatenation-based models, and a bunch of trainable bag-of-freebies designs highly optimized for the MS COCO dataset. We also provide the comparison results on Open Image V4 to show that yolo V6-L, V7 cannot beat our model in a more complex dataset which contains more dense scenario images with small objects and difficult background.

## D.2    Balancing Hyper-parameter Search

$\lambda$ is changed dynamically. In the early stage, it is set to be high ($\lambda = 1$) so the focus is on exploring the unknown patches. As training progresses, it decreases as $\lambda = \left(1 - \frac{N_c}{N_{epoch}}\right)$, where $N_c$ is the current epoch. Exact exploration-exploitation balancing also depends on complexity of dataset.

Table 7: SOTA YOLO baseline comparison on MS COCO test-dev and Open Image V4 test set

| Method | MS COCO | | | | | Open Image V4 | | | | |
|---|---|---|---|---|---|---|---|---|---|---|
| | AP | $AP^S$ | $AP^M$ | $AP^L$ | $AP^{CH}$ | AP | $AP^S$ | $AP^M$ | $AP^L$ | $AP^{CH}$ |
| yoloV5-L | 45.8 | 26.2 | 48.5 | 54.2 | 26.6 | 39.7 | 24.1 | 43.8 | 47.3 | 24.5 |
| yoloX-L | 46.9 | 26.5 | 49.1 | 55.4 | 27.2 | 40.5 | 24.6 | 44.3 | 48.0 | 25.1 |
| yoloE-L | 47.5 | 26.9 | 49.7 | 55.9 | 27.4 | 42.9 | 25.2 | 44.9 | 48.7 | 25.5 |
| yoloV6-S | 40.3 | 24.5 | 46.5 | 53.5 | 24.3 | 39.5 | 22.3 | 41.9 | 45.1 | 21.3 |
| yoloV6-M | 43.5 | 26.8 | 48.9 | 55.5 | 25.5 | 41.8 | 24.9 | 44.2 | 47.8 | 23.9 |
| yoloV6-L | 49.5 | 29.1 | 51.2 | 57.4 | 28.6 | 44.5 | 27.5 | 46.5 | 50.9 | 26.7 |
| yoloV7 | **49.8**±0.54 | 29.5±0.58 | **51.4**±0.52 | **58.2**±0.53 | 28.8±0.56 | 44.9±0.55 | 28.2±0.59 | 47.2±0.57 | 51.6±0.54 | 27.4±0.61 |
| AIRS | 48.3±0.58 | **32.1**±0.62 | 48.5±0.55 | 54.3±0.56 | **29.4**±0.63 | **47.5**±0.60 | **31.5**±0.65 | **48.1**±0.58 | **53.1**±0.58 | **29.0**±0.64 |

For instance, for easy dataset, the model may quickly focus on the exploitation part as epistemic uncertainty may reduce quickly whereas for difficult dataset, the model may stay longer to explore the patches. We also conduct an additional experiment to test sensitivity of $\lambda$. As shown in the Table 8, the performance is relatively robust for different $\lambda$ values. However, the adaptive $\lambda$ achieves a better performance.

Table 8: Impact of hyper-parameter $\lambda$

| Hyper-parameter $\lambda$ | COCO AP |
|---|---|
| 1 | 46.8 |
| 0.8 | 46.7 |
| 0.6 | 46.9 |
| 0.4 | 46.5 |
| 0.2 | 46.1 |
| **1→0 (AIRS)** | **48.3** |

### D.3 Datasets with a Large Amount of Small Objects

We clarify that COCO, Pascal VOC, and Open Images V4 are commonly used benchmark datasets to evaluate dense object detection models such as GFocal, DINO, FCOS etc. Therefore, we choose same set of datasets in our evaluation. To explicitly show the effectiveness of our technique, we further create a challenging subset, where large, medium and small objects are mixed and embedded with each other. This mixing strategy makes the detection highly challenging because such scenarios require a good balance of exploitation and exploration in RL training to achieve high precision and recall for all large, medium and small objects. Second, following the reviewer's suggestion for using datasets with more smaller objects, we redefine our criteria to select subsets that contain those images where the ratio of large and medium objects (area $\geq$ 322) to small objects (area $<$ 322) $\leq$ 1/2. We additionally conduct experiments on an aerial park lot dataset with a large amount of small objects in each image [16]. The quantitative results on the new challenging subset and an aerial dataset are summarized in Table 9. We also provide detection visualization of these two new challenge data sets in Figure 8.

Table 9: AP performance of AIRS on the aerial parking lot dataset [16] and newly created MSCOCO challenging subset, both containing a large number of cluttered small objects in one image, besides few medium or large objects.

| Data Set | GFocal | | | | AIRS | | | |
|---|---|---|---|---|---|---|---|---|
| | AP | $AP^S$ | $AP^M$ | $AP^L$ | AP | $AP^S$ | $AP^M$ | $AP^L$ |
| Aerial parking lot | 47.8 | 48.3 | 31.4 | 30.8 | 50.9 | 51.5 | 31.5 | 40.0 |
| New challenging subset | 32.5 | 33.8 | 30.4 | 31.9 | 33.2 | 34.6 | 30.3 | 31.9 |

### D.4 RL Agent Training Configurations

As compared with existing Q-learning models, our model is less computationally expensive to train due to two reasons: 1) The maximum time step $T$ is around 60 and in most cases the reward is positive (between 0 and 1), so it does not suffer from reward vanishing. 2) Training samples in our datasets are sufficient to train the RL agent. Given such potential advantages for RL training brought by AIRS, a Double-DQN target network is sufficient to stabilize the training. To further guarantee the training success and avoid early termination, we don't allow model to move upwards in Layer $L, L-1$ in the first 40 time steps of each RL training episode. Figures 9a, 9b, and 9c show the Q-learning loss with respect to training epochs, where the loss gradually decreases towards convergence. Furthermore, Figures 9d, 9e, and 9f show the average cumulative rewards, which exhibit a non-decreasing trend over all three datasets. This justifies that our approach achieves stable model training and the minimization in the loss which is also reflected by the cumulative reward.

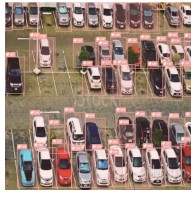 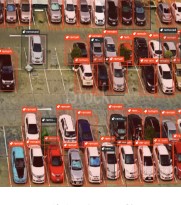 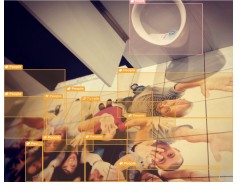 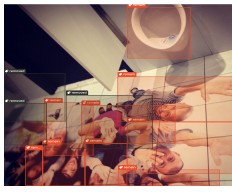

|        (a) GFocal        |        (b) AIRS        |        (c) FGocal        |        (d) AIRS        |

Figure 8: Results w/ and w/o AIRS generated masks on an aerial dataset [16] and a new challenging MSCOCO subset, both of which contain many small objects: (a) and (c) are the GFocal detection results while (b) and (d) are the results from AIRS after applying the RL masks. As can be seen, those grey boxes are the false positive bounding boxes, most of which possess irrelevant backgrounds or partial objects, masked out by the RL agent (i.e., 0 masks), while the red boxes are the remaining true positive boxes, which are kept the by the RL agent (i.e., 1 masks).

Table 10: Ablation on RL masks trained on Different Detector Backbones

| Method | MS COCO | | | | |
|---|---|---|---|---|---|
| | AP | $AP^S$ | $AP^M$ | $AP^L$ | $AP^{CH}$ |
| RetinaNet [26] | 39.1 | 21.8 | 42.7 | 50.2 | 21.6 |
| RetinaNet-RL | 42.5 | 25.0 | 41.8 | 50.2 | 25.8 |
| RetinaNet-Trans | 41.8 | 24.6 | 38.8 | 50.0 | 23.7 |
| FCOS [38] | 41.5 | 24.4 | 44.8 | 51.6 | 23.5 |
| FCOS-RL | 43.8 | 26.9 | 43.2 | 48.3 | 27.1 |
| FCOS-Trans | 43.2 | 26.4 | 42.7 | 47.6 | 25.8 |
| ATSS [45] | 43.6 | 26.1 | 47.0 | 53.6 | 23.8 |
| ATSS-RL | 45.5 | 29.5 | 45.7 | 52.2 | 27.9 |
| ATSS-Trans | 44.8 | 27.6 | 45.1 | 49.5 | 26.2 |

## D.5 Ablation Study on Base Detectors

To study the impact of base detectors, we select a set of representative one-stage detectors, including RetinaNet, FCOS and ATSS. For a fair comparison, we use ResNet-50 as the common backbone for all the methods. We first train our RL agent on the FPN of these pre-trained one-stage detectors. After training, we run the RL agent of each base detector and predict the binary masks for every test image. Finally, the RL masks are used to obtain the RL-augmented prediction results by training the underlying base detector again on the precise RL designated anchors. We apply the 'RL' suffix to these results. Meanwhile, we directly apply the RL masks generated from AIRS to the other detectors training phase and apply the 'Trans' suffix to these results. As Figure 10 shows, the RL masks independently trained on each base detector further improve the $AP, AP^S, AP^{CH}$ performance comparing to original baseline without any RL augmentations. Furthermore, we observe that the transferred RL masks trained from AIRS could also achieve performance improvement. We attribute such good transfer performance to the similar feature maps in the pre-train FPNs across different one-stage detectors.

## D.6 Train efficiency

The training time of AIRS is around 12 hours, which is far less than those one-stage detector's training time: 23h (RetinaNet), 24h (FCOS), 27h (ATSS), 27h (GFocal) with one A100 GPU of memory 40G. It is worth to note that we move the RL training to the pre-computing phase and apply RL learned masks onto the pre-trained FPN of GFocal to get the inference detection results. The comparison of AIRS training cost with other RL baselines is provided in Table 11.

Table 11: RL baseline training efficiency comparison

| Model | Training Time (hours) |
|---|---|
| Hierarchical-RL | 14 |
| Caicedo-RL | 16 |
| Tree-RL | 23 |
| Multiple-RL | 35 |
| ReinforceNet | 33.4 |
| AIRS | **12** |

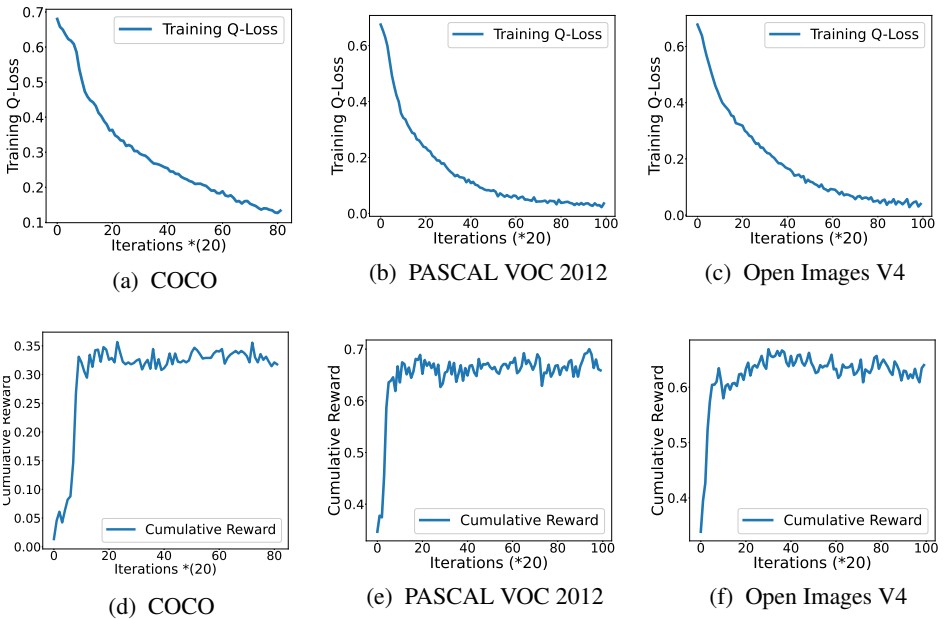

Figure 9: Q-learning loss (top row) and cumulative reward (bottom row) of the RL agent

### D.7 Subset Generation Illustration

Figure 10 provides additional examples to compare the detection results from GFocal v.s. `AIRS` on the subsets of three public datasets, respectively. As can be seen, `AIRS` is highly effective in reducing false positive detections as compared to one of the most competitive baselines, GFocal. A close look at the example images reveals that most false positive detections come from small duplicate bounding boxes that focus on local texture, which is usually unnecessary.

## E    Limitation and Broader Impact

We identify two additional limitations of the proposed `AIRS` model. First, the evidential Q-value integrates the epistemic uncertainty predicted by the model. If the model is poorly calibrated, its uncertainty quantification becomes less trustworthy that can negatively impacts the exploration effectiveness of the RL agent. We plan to investigate effective network calibration methods and integrate them into the proposed model. The second limitation is that `AIRS` relies on the feature pyramid network (FPN), which leverages ResNet as its backbones. A systematic extension to the transformer based backbone may have the potential to further improve the detection performance.

`AIRS` is designed to be integrated and applied on top of one-stage detector's FPNs for object detection. For other domains, we believe there is potential for similar technique to be applied on them as well. For example, video tracking requires accurate object localization, which could be achieved by defining large-medium-small image patch hierarchy and conducting object detection on each patch to collect reward. Note that it is different from ours because we conduct a hierarchical patch search on pre-trained FPN, and then regress bounding boxes for every positive anchor on top of the selected patches. Leveraging FPN guarantees a high recall, which helps in dense scenarios while not in (single) accurate object detection. As for segmentation, it is a pixel-level classification, how to select foreground pixels (positive anchors) residing in interested object area (patches) could be achieved by one-stage detector's FPN with `AIRS` augmentation.

## F    Link to Source Code

For the source code, please check `https://github.com/ritmininglab/AIRS`.

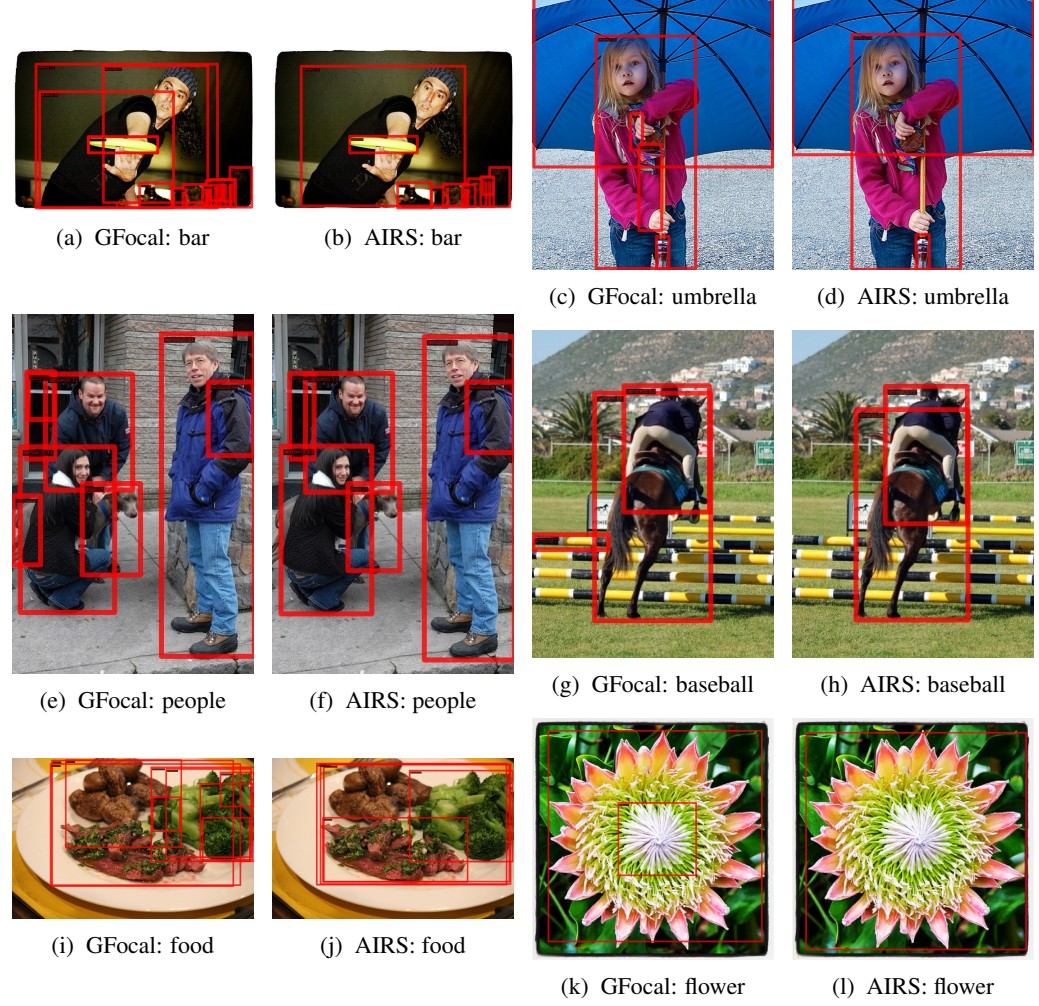

(a) GFocal: bar      (b) AIRS: bar

(c) GFocal: umbrella      (d) AIRS: umbrella

(e) GFocal: people      (f) AIRS: people

(g) GFocal: baseball      (h) AIRS: baseball

(i) GFocal: food      (j) AIRS: food

(k) GFocal: flower      (l) AIRS: flower

Figure 10: Qualitative comparisons between GFocal (Left) and AIRS (Right) detection results on three datasets: COCO 10a-10d, PASCAL VOC 2012 10e-10h and OpenImages V4 10i-10l

