# OpenReview forum: "Adaptive Important Region Selection with Reinforced Hierarchical Search for Dense Object Detection"
_NeurIPS.cc/2024/Conference — NeurIPS 2024 poster_

### Official Review · Reviewer_A3eZ · 2024-07-12

**Soundness:** 3
**Presentation:** 3
**Contribution:** 3
**Rating:** 7
**Confidence:** 3

**Summary:**

The paper presents a novel RL-driven object detector guided by Evidential Q-learning. The main contributions are:
1. an adaptive hierarchical object detection paradigm supported by an RL agent to mimic human visual attention that performs searching in the top-down fashion;
2. an evidential Q-learning method driven by a unique reward function, covering both potentially positive and highly uncertain patches
3. theoretical guarantee on the fast convergence of the proposed evidential Q-learning algorithm.

Experimental results show the effectiveness of their method.

**Strengths:**

1. The motivation and the proposed method are interesting and technically sound.
2. Most part of the manuscript is clearly written and easy to understand.
3. Enough ablation study experiments to show the effectiveness of the proposed method.

**Weaknesses:**

1. This paper does not include comparison with the latest works in dense object detection , as the proposed modules are simply tested using a few weakness baselines. I am wondering if those techniques can be used to improve accuracy of state-of-the- art models? For example, [1] [2] .

[1] Xu D, Deng J, Li W. Revisiting ap loss for dense object detection: Adaptive ranking pair selection[C]//Proceedings of the IEEE/CVF Conference on Computer Vision and Pattern Recognition. 2022: 14187-14196.

[2] Hou X, Liu M, Zhang S, et al. Salience DETR: Enhancing Detection Transformer with Hierarchical Salience Filtering Refinement[C]//Proceedings of the IEEE/CVF Conference on Computer Vision and Pattern Recognition. 2024: 17574-17583.

2. My concern is whether the RL baseline comparisons should also be included in the main experiments (Table 1 and Table 2)..

**Questions:**

See Weaknesses.

**Limitations:**

The authors discussed limitations thoroughly.

---

> ### Author Rebuttal · Authors · 2024-08-07
>
> **Q1: Performance comparison with [1] [2]**
>
> Thank you for providing the references for these baselines. We would like to clarify that we focus primarily on improving the dense object detection performance by effectively discovering all objects (including the smaller ones) through leveraging the FPN structure and filtering out false positive cases using our novel evidence guided exploration-exploitation strategy coupled with adaptive hierarchical searching. The first baseline uses a novel adaptive Pairwise Error (APE) loss that focusing on ranking pairs in both positive and negative samples. The second is more like a refinement built upon the DETR model. It proposes hierarchical salience filtering refinement, which performs transformer encoding only on filtered discriminative queries, for a better trade-off between computational efficiency and precision. As such, both methods don't leverage the FPN structure for soliciting enough positive anchors as the candidate pool and may suffer in the dense scenario, where many smaller objects coexist with, overlap with or are included in the large objects. As experimental evidence shown in table below, both methods' AP performance is worse than our proposed technique, especially for small objects $AP^{S}$. We conduct comparison experiment with MSCOCO data set and use the same backbone ResNet-50-FPN and epoch number in the first baseline for a fair comparison. For the second baseline which is orthogonal to ours, we re-run their pre-trained model to get the reported performance.
>
>
> | **Model**           | **AP** | **$AP^{S}$** | **$AP^{M}$** | **$AP^{L}$** | **$AP^{CH}$** |
> |-----------------|------------|-----------|-----------|-----------|-----------|
> | Adaptive Pairwise Error | 41.5 | 23.5 | 45.7 | 52.6 | 22.4 |
> | Salience DETR | 46.5 | 15.4 | 46.8 | 53.5 | 15.5 |
> | AIRS | 47.6 | 31.0 | 48.5 | 54.3 | 30.2 |
>
> **Q2: RL Baseline Comparisons in Table 1 and Table 2**
>
> Thank you for the suggestion! We will add selected RL baselines as part of Tables 1 and 2 in the revised paper.

---

> > ### Author Response · Authors · 2024-08-12
> >
> > Dear Reviewer A3eZ,
> >
> > Thank you again for your constructive comments and insightful questions! In the rebuttal, we have provided additional experimental result on mentioned baselines [1, 2]. We would also like to acknowledge that we will include RL baselines as part of Tables 1  and 2 in the revised paper.  By addressing your comments, we believe that quality of the paper has been improved and we appreciate the reviewer's support on that. We hope that reviewer finds our answers satisfactory! We are more happy to answer any additional questions that you may have.

---

> > > ### Comment · Reviewer_A3eZ · 2024-08-13
> > >
> > > My concerns are addressed and I intend to keep the original rating.

---

> > > > ### Author Response · Authors · 2024-08-13
> > > >
> > > > We appreciate your confirmation that our rebuttal has addressed your concerns. Thank you for keeping the positive rating on the paper!

---

### Official Review · Reviewer_Q21q · 2024-07-13

**Soundness:** 3
**Presentation:** 3
**Contribution:** 3
**Rating:** 6
**Confidence:** 3

**Summary:**

The paper presents an innovative framework for dense object detection, called Adaptive Important Region Selection (AIRS). It introduces a method guided by Evidential Q-learning, which strategically identifies important regions within an image in a hierarchical manner. The method aims to reduce false positives commonly produced by current dense object detection techniques by dynamically balancing exploration and exploitation during the model training phase.

**Strengths:**

1.	Introducing Evidential Q-learning into the hierarchical selection process for object detection is novel.
2.	The paper is well-written, with a clear motivation and a well-defined methodology. The theoretical analysis is comprehensive.
3.	Extensive experimental validation across multiple datasets shows the framework's effectiveness against state-of-the-art techniques.

**Weaknesses:**

1.	It would be beneficial to visualize and analyze some intermediate results, such as the RL masks in the test phase.
2.	The proposed AIRS involves searching for region masks before making predictions, aligning more closely with two-stage approaches. Conversely, the DETR series of models directly predict bounding boxes in an end-to-end manner, eliminating the need to obtain candidate regions beforehand. This characteristic classifies them as one-stage methods.
3.	The citation for DINO seems incorrect; reference [21] actually cites DN-DETR.

**Questions:**

1.	How does AIRS handle extremely cluttered scenes where objects are not only dense but also partially occluded?
2.	Can you explain why AIRS significantly underperforms in detecting large objects in the COCO dataset compared to DINO, yet shows better performance on the other two datasets?

**Limitations:**

The authors have adequately discussed the limitations in terms of the potential negative impacts of low-quality uncertainty quantification and the challenges of extending to a transformer-based backbone.

---

> ### Author Rebuttal · Authors · 2024-08-07
>
> **Q1: Intermediate results of RL masks during testing phase**
>
> Thank you for the suggestion. Figure 1 in the attached PDF shows the RL masks that are projected to the removed  false positive bounding boxes in the detection result. For more detailed description of the masking process, please refer to the answer to **Q2** in the General Response.
>
>
> **Q2: The proposed AIRS involves searching for region masks before making predictions, aligning more closely with two-stage approaches. Conversely, the DETR series of models directly predict bounding boxes in an end-to-end manner, eliminating the need to obtain candidate regions beforehand. This characteristic classifies them as one-stage methods.**
>
> Thank you for the insightful comment. First of all, we would like to clarify that we do not claim our approach to be One-stage detector. As our technique leverages the FPN which is indeed one-stage detector, therefore our technique is based on one-stage detector. We agree with the reviewer that our technique can be interpreted as a two stage detector as we perform the RL training and inference on the top of one-stage FPN pre-trained network.
>
> Regarding, DEtection TRansformer or DETR, we believe that it leverages a set-based global loss that forces unique predictions via bipartite matching and it leverages a transformer encoder-decoder architecture for that. Given a fixed small set of learned object queries, DETR reasons about the relations of the objects and the global image context in order to directly output
> the final set of predictions in parallel. As such, our understanding is that it is different from existing two-stage detectors (e.g., RPN) as well as one-stage detector (FPN) as it does not rely on anchor generation, selective search and post-processing. However, considering its nature, we can regard it more closely related to the one-stage detector rather than two-stage detector. We will make this clear in the revised paper.
>
>
> **Q3: DINO Citation.**
>
> Thanks for pointing our this typo. We will correct the citation as DN-DETR in our revised paper.
>
>
> **Q4: Handling partially occluded in extremely cluttered scenes**
>
> Please refer to the answer to **Q4** in the General Response.
>
>
> **Q5: AIRS inferior performance on large objects in MS COCO**
>
> Please refer to the answer to **Q1** in the General response.

---

> > ### Author Response · Authors · 2024-08-12
> >
> > Dear Reviewer Q21q,
> >
> > Thank you again for your constructive comments and insightful questions! In the rebuttal, we have
> >
> > - provided intermediate results of RL masks in the testing phase (Please refer to Figure 1 in the attached PDF),
> >
> > - further compared the difference between AIRS and DETR,
> >
> > - explained the relatively lower performance of AIRS on MS COCO large objects,
> >
> > - justified how our exploration-exploitation strategy effectively removes the smaller partially occluded objects in extremely cluttered scenes (Please refer to the answer to Q4 in the General Response).
> >
> >
> > By addressing your comments, we believe that quality of the paper has been improved and we appreciate the reviewer's support on that. We hope that reviewer finds our answers satisfactory! We are more happy to answer any additional questions that you may have.

---

> > > ### Comment · Reviewer_Q21q · 2024-08-12
> > > **Respond to the authors**
> > >
> > > The authors have addressed most of my concerns, and I keep my original rating.

---

> > > > ### Author Response · Authors · 2024-08-13
> > > >
> > > > Many thanks for reading our rebuttal and confirming that it has addressed most of your concerns. We appreciate that you maintain the positive rating on the paper!

---

### Official Review · Reviewer_mhHL · 2024-07-19

**Soundness:** 2
**Presentation:** 3
**Contribution:** 2
**Rating:** 5
**Confidence:** 4

**Summary:**

This article presents the method AIRS (Adaptive Important Region Selection) based on reinforcement learning paradigm to improve the performance of dense object detection in images.
It is highlighted that best SOTA object detectors either provide too many false positive detections in complex scenes, or fail at proposing positive candidates to detect all small objects.
The proposed method aims at searching for image patches containing detections in a top-down hierarchical (multi-scale) fashion. To this aim, it balances between exploration of unknown patches (by using evidential Q-learning to encode epistemic uncertainty) and exploitation of patches containing detections.
A theoretical analysis of fast convergence of the proposed algorithm is given. The method is evaluated on 3 object detection datasets, compared with SOTA models. An ablation study about backbones and loss choice is given.

**Strengths:**

- The limitation of current object detectors in presence of possibly small objects in dense and complex scenes is a problem of utmost importance in the computer vision domain. Solutions to the problem would be very useful for many applications.

- The method seems original to me, with a good idea of using RL to explore and exploit the different regions of the image for potential presence of objects.

- A theoretical analysis is provided with the proposed RL method.

- The results outperform SOTA methods.

- In general, the paper is well written and clear.

**Weaknesses:**

- My main concern is about the general claim of better managing small objects and dense complex scenes. It is not clearly demonstrated.
The datasets used are quite general to evaluate object detection. The most complex is OpenImages with about 8.4 objects per image, which is not so dense.
To demonstrate the effect of the proposed method, it would be better to use other datasets like aerial/satellite datasets with a big amount of small objects in each image.
Even if the method seems interesting and overall well-performing, it is not clear that it is more efficient for dense scenes or small objects.

- Besides, the subset for the $AP_{CH}$ computation does not separate the most challenging images considering these aspects. Evaluation should be done on other subsets of challenging cases.

- Some poor results about large objects are unexpected. Why does the exploration/exploitation mecanism decrease the performance in detecting such objects? Does the method focus too much on objects of smaller scales?

- The role and effect of hyper-parameter $\lambda$ is not studied. However, it seems to be important.

- The ablation study for choosing DIoU instead of GIoU is not convincing. Conclusions should have been the opposite.

- Some quality and clarity check should improve the paper, as detailed in the following section.

**Questions:**

- As dense object detection and, specifically, small object detection are the goals of the proposed method, why only general object detection datasets were used for evaluation?
PascalVOC have mostly images with a unique big object, MSCOCO images have a bit more objects per image and at most, OpenImage images have only 8 objects/images on average.
To draw conclusions about the capacity of AIRS to detect smaller objects and better detect all objects in images with a high density of object, other datasets should be used for evaluation (e.g. the many aerial images datasets like DOTA, xview...).

- Regarding the performance results on MS COCO (Table1), how do you explain AIRS has poor performance in detecting large objects (54.3% vs 62.5% for best SOTA) and only outperforming FasterRCNN?
For medium object, it is not so bad but still does not outperform SOTA (48.5% vs 50.4%).
For challenging subset, it is slightly better than SOTA (+0.4pp).
For small objects, it outperforms SOTA by +2pp.
Thus, it seems the AIRS has specialized in smaller objects but is less effective for bigger objects in the MS COCO. It is rather unexpectable if we compare with the other datasets VOC and OpenImages.
What would be your explanation of this unexpected poor performance?

- Is there a link between the top-down hierachical strategy of AIRS and the cases of small object boxes that are fully contained in other bigger ones?
How does the top-down method handle these cases? Is it observable on a challenging subset dedicated to "included boxes"?

- Hyper-parameter $\lambda$ (line 156) seems to be important for a good balance. How to choose it? Do you have any sensitiviy study about it? It would be interesting to see how $\lambda$ actually affects the final performance on the different types of objects through the exploration-exploitation trade-off.

- The definition of challenging subset (line 273) is surprising. Why using only the images with ratio of large and medium over small objects ranging from 1 to 1/2? Why not simply <1 to have complex scenes with both large and tiny objects?
Criterion (b) is not clear. What is the measure and threshold of overlap used to select the images?
Criterion (c) is not clear. What is the measure and threshold of object inclusion used to select the images?
It is not clear if the union or intersection of these criteria is used to define the subset.
It would be interesting to separate in several subsets in order to draw more precise conclusions about the ability of the proposed detector in managing each case better than the SOTA methods.

- "DIoU is the most effective" (l.355-356). The results from Tab.4 show exactly the contrary. DIoU is the less effective. According to Tab.4, GIoU is the best choice. So, logically, I would expect AIRS combine GIoU and Uncertainty. Why didn't you use this combination? This result should be added in the ablation study.


Here are some other comments, typos or lack of clarity in some phrasing that should be corrected or clarified:
- All references to section/equation/... should have the word Section/Equation before the number. (e.g. lines 43, 62, 64, 115, 117...). Please check them all.
- l.22: "The diverse nature of images, such as shadow/occlusion" sounds awkward. Please rephrase it.
- l.29: "number of candidate object". typo.
- l.36: "inconsistency in localization quality estimation between training and testing". The idea behind this statement is not clear.
- Fig.1: The same image should be used for the qualitative comparison of the 4 methods.
- Fig.1: The name of the method GFocal+LQE in the subcaption c) should match with the name in the caption.
- l.42: "generating too many false positive predictions on small objects" It is not clear in this sentence whether the problem is about duplicated predictions on small objects (then, some post-processing, like NMS, can leverage this issue) or about false positive predictions on small areas (which are not objects).
- l.96: "leverages the latest Feature Pyramid Network structure" Please clarify the meaning of 'latest' (FPN was proposed at CVPR'2017).
- l.110: "each of the key component". typo.
- l.121: Please define the acronym NIG.
- Fig.2: Please add the layer scale (0 to L-1) on the diagram for better link with the main text.
- l.135-l.143: Please avoid using the same variable d for two different usages.
- l.140-160: Please define all variables (e.g. $\alpha, \beta, \gamma, \nu$)
- Eq.7: Please define $\gamma$.
- Eq.8, l.201: Typo $n_{epcoh}$
- l.242: "leveraging the FPN structure of the pre-trained backbones." Which type of pre-training was used? Is it the same for all compared methods? What is its influence on the performance results?
- l.251: "As can be seen, the training is more efficient comparing to other RL based methods, and the inference speed is also competitive w.r.t. the latest baselines (see Appendix D.5)." Please be more specific than "more" (adding some figures illustrating these statements).
- l.266: "It contains 20 categories partitioned into three subsets" Please rephrase as it appears the categories are partitioned.
- l.268: typo
- l.272: CH subset should be made available for comparison with future work in the community.
- l.274: add unit
- l.275: typo
- l.286: Detail which supervision and dataset were used for the pre-trained models, and if it is identical to compared methods.
- l.289: Please detail the bounds of search.
- l.292: "We gradually shrink λ" Please clarify if it is done for the hyper-parameter search only or during the training also? Please be more specific on the shrinkage process for reproducibility purpose.
- l.294: Which criteria were exactly used to stop the training?
- l.296: SOTA methods are not so recent, exept Co-DETR and EVA (2023). Have you checked more recent detectors?
- Tab.1: Add % for AP results in all tables.
- I was curious about the training time (then found it in the appendix). It would be beneficial to add a synthetic sentence about it in the main paper.
- Table2: Please specify which scale version of DINO was used.
- Table2: Please add the resnet variants in the same table for comparison.
- l.309: Please specify which pre-training was used.
- l.312: A short sentence summing up the results on the YOLO series should be added, even if all the details are in appendix.
- l.322: Please use pp (percentage points) instead of %.
- l.340-341: Please rephrase.
- Fig.3: a)b) Please correct the vertical axis name (number of...)
- Fig.3: Please add the statistics of the ground truth for a complete comparison.
- Section4.5: It is awkward to use MSCOCO for the ablation study, as AIRS gives unexpected results for medium/large objects.
- l.364: Please rephrase.
- l.370-371: Please define the acronym EL (only EU was defined).

**Limitations:**

As detailed before, evaluation on relevant datasets are missing to support the main claim of dense and small object detection capacity.

---

> ### Author Rebuttal · Authors · 2024-08-07
>
> **Q1: Use other datasets with a big amount of small objects.**
>
> Thank you for the great suggestion! First, we would clarify that COCO, Pascal VOC, and Open Images V4 are commonly used benchmark datasets to evaluate dense object detection models such as GFocal, DINO, FCOS etc. Therefore, we choose same set of datasets in our evaluation. To explicitly show the effectiveness of our technique, we further create a challenging subset, where large, medium and small objects are mixed and embedded with each other. This mixing strategy makes the detection highly challenging because such scenarios require a good balance of exploitation and exploration in RL training to achieve high precision and recall for all large, medium and small objects. Second, following the reviewer's suggestion for using datasets with more smaller objects, we redefine our criteria to select subsets that contain those images where the ratio of large and medium objects (area $\ge$ 322) to small objects (area $<$ 322) $\leq$ 1/2. We additionally conduct experiments on an aerial park lot dataset with a large amount of small objects in each image. The quantitative results on the new challenging subset and an aerial dataset are summarized in Table 1 in the attached PDF. We also provide detection visualization of these two new challenge data sets in Figure 1 of the attached PDF.
>
>
> **Q2: Poor performance on large objects and impact of  exploration/exploitation**
>
> Please refer to the answer to **Q1** in the General Response. In fact, the lower performance on large object setting of MSCOCO primarily comes from the one-stage detectors and the nature of the MSCOCO dataset not because of our exploration-exploitation technique. Due to exploration/exploitation mechanism, our approach effectively discovers small objects in addition to the larger ones. However, if our approach focuses too much on small objects, then we would consistently see the lower larger object performance across all datasets. The comparable or even better performance on other datasets except for MS COCO further justifies the effectiveness of the exploration/exploitation mechanism.
>
>
>
>
> **Q3: Handle small objects embedded within large objects cases and challenging subset dedicated to "included boxes"?**
>
> For this first part of the question, please refer to our answer to  **Q4** in the General Response. Regarding the second part, we would like to clarify that our challenging dataset measured by $AP^{CH}$ already covers these situations through criterion c, which is also illustrated in Figure 9 of Appendix D.6.
>
> **Q4: Hyper-parameter $\lambda$**
>
>  $\lambda$ is changed dynamically. In the early stage, it is set  to be high ($\lambda = 1$) so the focus is on exploring the unknown patches. As training progresses, it decreases as $\lambda = \left(1-\frac{N_c}{N_{epoch}}\right)$, where $N_c$ is the current epoch.  Exact exploration-exploitation balancing also depends on complexity of dataset. For instance, for easy dataset, the model may quickly focus on the exploitation part as epistemic uncertainty may reduce quickly whereas for difficult dataset, the model may stay longer to explore the patches. We also conduct an additional experiment to test sensitivity of $\lambda$. As shown in the table below, the performance is relatively robust for different $\lambda$ values. However, the adaptive $\lambda$ achieves a better performance.
>
> | **Hyper-parameter $\lambda$**           | **COCO AP** |
> |-----------------|-----------------------|
> | 1 | 46.8                  |
> | 0.8      | 46.7                    |
> | 0.6         | 46.9                   |
> | 0.4     | 46.5                   |
> | 0.2    | 46.1                 |
> | **AIRS**   | **48.3**              |
>
>
>
>
>
> **Q5: The definition of challenging subset.**
>
> In case of criteria (a), we consider images where the ratio of larger/medium objects (area $\geq$ 322) to small objects (area $<$ 322) ranging from 1 to 1/2 to ensure all sized objects coexist. This mixing strategy makes the detection really challenging because such scenarios require a good balance to achieve high precision and recall for all objects. We did not consider the ratio less than 1/2 because the selected objects are very small ones that are even hard for human to detect. Following the suggestion, we conducted an additional experiment by considering objects where ratio is less than 1/2. Please refer to our answer to **Q1** for the results. In case of (b), we consider the case where images with multiple objects overlap with each other. In case of (c), we consider the case where multiple small objects are embedded into a bigger one. In both cases, we consider the threshold of 0.4. This means, in case of (b), if the IOU between overlapping objects is more than 0.4 then we consider those samples. In case of (c), if the IOU of smaller objects with the bigger ones is higher than 0.4, we consider those samples. Apart from the IOU threshold, in both cases, in order for given image to be qualified for (b) or (c), the minimum number of small objects should be at least 3.
>
>
>
> **Q6: DIoU vs. GIoU**
>
> Thanks for pointing out this inconsistency. We re-checked our experiment results and found that row numbers are shifted in the paper. Below, we include GIoU with uncertainty performance and will update it accordingly in revised paper.
>
>
> | **Model Design Choice**           | **AP** | **$AP^{S}$** | **$AP^{M}$** | **$AP^{L}$** | **$AP^{CH}$** |
> |-----------------|------------|-----------|-----------|-----------|-----------|
> | GIoU | 44.3 | 28.5 | 45.6 | 51.5 | 25.4 |
> | GIoU+Uncertainty | 46.7 | 30.2 | 47.5 | 53.4 | 28.1 |
> | DIoU | 45.4 | 29.5 | 46.8 | 52.4 | 26.7 |
> | DIoU+Uncertainty | 47.6 | 31.0 | 48.5 | 54.3 | 28.9 |
>
>
>
> **Q7. Typos or lack of clarity**
>
> Thanks for carefully checking our paper! We highly appreciate your suggestions and will follow them to fix the issues and improve the presentation of the revised paper.

---

> > ### Author Response · Authors · 2024-08-12
> >
> > Dear Reviewer mhHL,
> >
> > Thank you again for your constructive comments and insightful questions! In the rebuttal, we have
> >
> > - conducted additional experiments
> > using other datasets with more small objects,
> >
> > - clarified the relatively low performance of AIRS on the MS COCO large objects,
> >
> > - discussed how our approach handles smaller objects embedded within large ones,
> >
> > - analyzed the sensitivity of the performance with respect to hyperparameter $\lambda$.
> >
> > We believe that by addressing your suggestions, our paper has been significantly strengthened and we appreciate the reviewer's support on that. We hope that reviewer finds our answers satisfactory and considers updating the score accordingly! We are more happy to answer any additional questions that you may have.

---

### Official Review · Reviewer_8nSu · 2024-07-21

**Soundness:** 3
**Presentation:** 3
**Contribution:** 2
**Rating:** 5
**Confidence:** 4

**Summary:**

Current state-of-the-art dense object detection techniques often generate numerous false positive detections in complex scenes, as they prioritize high recall. This study tackles this problem by introducing an Adaptive Important Region Selection (AIRS) framework. This framework builds on a pre-trained FPN-based detector and uses evidential Q-learning to identify the most informative patches from the top layer to the bottom layers during training and testing. To enhance performance, the authors propose a uniquely designed reward function based on diverse detection metrics. Experiments on three standard detection benchmarks demonstrate the effectiveness of the proposed AIRS. Additionally, the authors provide a theoretical analysis of AIRS, which helps readers better understand the method.

**Strengths:**

1. The paper is generally easy-to-follow.

2. The proposed method makes interesting use of Q-learning to select important regions to retrain an object detector.

3. The experiments cover a wide range of different datasets

4. The method displays good performance across most of evaluation metrics over state-of-the-art methods.

5. The authors consider a number of different ablations to better understand the proposed method.

**Weaknesses:**

1. One of my most important concerns is that AIRS should be used along with a well-trained detector. In this study, the authors use an detector, equipped with FPN and pre-trained by GFocal. As a result, it is not an easy-to-use framework, and it is also unfair to compare the training time with other end-to-end methods in D.5. Does AIRS can be used for end-to-end training of a dense object detector? How much time will be cost by the end-to-end training?

2. My second concern is that AIRS is similar to the region proposal network, but uses a different network (RNN v.s. MLP), training strategy (RL v.s. Standard Training) to select important regions for detection. However, RPN can be integrated into a detector to form a 2-stage detector. In fact, the masked region step can also be treated as a method of proposing regions. Therefore, it should compare AIRS and RPN in depth in this study, including presentation, experiments, etc. As I see, changing several hyper-parameters in RPN can also produce high recall as the main claim in AIRS.

3. I'm also wondering what kind of insights this work could bring to the community. I will be more interested in seeing which kinds of patches (locations) should be used for training and testing. Are patches used similar among different detectors (different training strategies, losses, backbones, etc.) ? Answering these questions would help the paper go beyond engineering success in standard detection benchmarks.

4. No limitation parts.

**Questions:**

See Weaknesses

**Limitations:**

There are no discernible negative societal impacts related to this work.

---

> ### Author Rebuttal · Authors · 2024-08-07
>
> **Q1: It is not an easy-to-use framework. Can AIRS be used for end-to-end training of dense detector and how much is the training time.**
>
> Thank you for the insightful question. We would like to clarify  that our goal is to remove the false positive (fp) bounding boxes, which is orthogonal to any FPN-based pre-trained detectors. As these pre-trained detectors are off-the-shelf and using them has been a common trend in the community, integration of our proposed technique with these detectors is straightforward, making our framework relatively easy to use.  Since our method works on top of the pre-trained object detectors to eliminate the false positive cases, we agree with the reviewer that the AIRS training time is not directly comparable to those underlying detectors. Instead, we can train them in an iterative way by further leveraging AIRS to fine-tune the detectors, which makes the whole training process end-to-end. The added training time of the detectors should be similar to those well established frameworks since we only add the RL filtering masks in the training. We would also like to clarify that the training time comparison shown in Table 9 is fair as all RL techniques start from the same the pre-trained  FPN environment.
>
>
> **Q2: Comparison with RPN**
>
> Please refer to the answer to **Q3** in General Response for the detailed discussion on the difference from RPN.
>
>
> **Q3: Changing several hyper-parameters in RPN can produce high recall**
>
> We would like to clarify that we have compared with a good number of two-stage detectors that leverage RPN (see the Two-stage section of Table 1). For each of these baselines, we consider the optimal set of hyperparameters resulting into the highest AP. Therefore, changing hyperparameter may not be able to further enhance the recall.
>
> **Q4: What kind of insights this work could bring to the community? Which kinds of patches (locations) should be used for training and testing? Similarity of RL selected patches for training and testing among different detectors (different training strategies, losses, backbones, etc.)**
>
> Thank you for these comments. The design of our adaptive important region selection with reinforced hierarchical search framework is inspired by the human visual attention, which usually conducts object search in a top-down hierarchical fashion. Such a mechanism makes the search very efficient as the RL agent moves down to a fine-grained level in the hierarchy only when it is likely to contain a object of interest. Furthermore, the epistemic uncertainty guided exploration plays a central role that ensures the RL agent will not miss any important regions (i.e., those with a high uncertainty) while avoiding visiting the unnecessary ones (i.e., those with a low uncertainty). Both our theoretical analysis and empirical results confirm the effectiveness of the proposed design strategy, which shows its potential to benefit similar dense object detection scenarios.
>
> Regarding which kinds of batches should be used, there is no explicit control on selection of the patches during training and testing phases. As our AIRS is based on FPN coupled with adaptive hierarchical search strategy guided by evidential Q-learning, it looks for informative patches that are likely to have objects in different granularity. As such, during training process, our RL agent learns to effectively find the patches containing objects of different sizes and and different types. This capability will transferred to the inference process. For the same image, RL selected patches display similar patterns across different backbone detectors. As shown in Table 8 in Appendix D.4, different pre-trained detectors (RetinaNet-Trans, FCOS-Trans, ATSS-Trans) with same GFocal generated RL mask and different RL agent generated masks (RetinaNet, RetinaNet-RL, RetinaNet-Trans) with the same backbone detector (RetinaNet) both display similar Average Precision (AP) on COCO in the inference step, which is a clear evidence that the RL selected patches generated from different agents and applied among different detectors should be similar and transferable to each other in some extent. Furthermore, we have provided an additional evidence of having similar patches among different agents and backbone detectors in Figure 2 of the attached PDF.
>
>
> **Q5: No limitation part**
>
> We have included a discussion of limitations in  Appendix E.

---

> > ### Author Response · Authors · 2024-08-12
> >
> > Dear Reviewer 8nSu,
> >
> > Thank you again for your constructive comments and insightful questions! In the rebuttal, we have
> >
> > - discussed how AIRS can be conveniently used and the training time if end-to-end training is performed along with the detectors,
> >
> > - compared with RPN by clarifying that RPN is less effective to detect small objects and why changing the hyper-parameters may not be able to further improve the performance,
> >
> > - discussed the insights that this work can bring to the community and how AIRS adaptively selects patches through the uncertainty-aware exploration-exploitation coupled with the adaptive top-down hierarchical search strategy,
> >
> > By addressing your comments, we believe that quality of the paper has been improved and we appreciate the reviewer's support on that. We hope that reviewer finds our answers satisfactory and considers updating the score accordingly! We are more happy to answer any additional questions that you may have.

---

### Official Review · Reviewer_4PiS · 2024-07-27

**Soundness:** 4
**Presentation:** 4
**Contribution:** 4
**Rating:** 8
**Confidence:** 2

**Summary:**

The paper presents an adaptive hierarchical object detection framework for dense object detection by evidential Q-learning with specially designed reward function, searching through FPN based hierarchy in a top-down fashion. Theoretical analysis proves the upper bound of the action value error and extensive experiments on various datasets illustrate the superiority of the proposed framework over other SOTA models.

**Strengths:**

1.  The paper proposes a novel evidential Q-learning method in an adaptive hierarcical top-down searching framwork for dense object detection
2.  The theoretical anlysis on the upper bound of the action value error is thorough, and guarantees the fast convergence of the Q-learning
3.  The experimental evaluations are extensive and thorough, the AP metric scores of the propose AIRS on different backbones are highest in most categories (except M&L in MS COCO) in all evaluated datasets, demonstrating the effectiveness of the proposed model.

**Weaknesses:**

1. The main intention of the proposed AIRS framework is to reduce the false positive cases in complex scenes. I would hope the author give clearer definition/description on what kind of "false positve" cases they want to avoid and provide visual examples to justify the claim. I see some ambiguity based on the current visual illustrations in the paper.  Using Fig 1(a), (c) and Fig 9(k) as examples, if I understand the proposed method correctly, the mask values for the smaller boxes are still 1, because they are still part of the elephant, banana and flower respectively, but it seems authors view them as "false positve".
2. For the experiment, especially qualitative comparisons, I would like to see more challenging examples. For current ones, I see most smaller unnecessary bounding boxes are fully within another bigger bounding box. In this case, a simple post processing is enough to remove these redundant small bounding boxes out from any exsiting models.

**Questions:**

1. Fig. 2 shows the overview of the AIRS framework and the RL inference step outputs the binary RL mask, but how the mask is used to generate the final dectected bounding boxes? This is not so clear to me, I hope the authors can explain more on the inference step.
2. How does the binary RL mask help reduce #unncessary bounding boxes?

**Limitations:**

Yes, the authors have addressed the limitations adqueately in the appendix section E.

---

> ### Author Rebuttal · Authors · 2024-08-07
>
> **Q1: Clearer definition/description on what kind of "false positive" cases authors want to avoid and provide challenging visual examples to justify the claim.**
>
> Thank you for the insightful question. A "false positive" detection covers two cases: i) a bounding box covering only part of an object, and ii) a bounding box covering irrelevant information (e.g., background  treated as elephant as shown in the right center of Figure 1(a)). In object detection, ideally we want to predict bounding boxes that completely cover all relevant objects. To this end, the training data contains ground truth bounding boxes that cover complete objects instead of partial ones. Bounding box capturing part of the object may be misleading.  For instance, in Figure 1(a), one of the bounding box captures the ear of the elephant, which is not precise to refer it as the whole elephant. This is why the false positive cases also include the partial bounding boxes. To more clearly demonstrate the effectiveness of our technique in better avoiding irrelevant information (i.e., case ii), we have performed an additional qualitative analysis. As shown in Figure 1 of attached PDF, our AIRS model successfully removes false positive cases that capture irrelevant background information.
>
> **Q2: A simple post-processing should be enough to remove redundant small bounding boxes.**
>
> Thank you for your thoughtful comment! We would like to clarify that the smaller bounding boxes are usually associated with a high confidence score. As such, the standard post-processing algorithms like Non Maximum Suppression (NMS) cannot properly remove them. This is also demonstrated in Figure 1 of the paper, where GFocal (which leverages NMS) is unable to remove the redundant bounding boxes. Further, evidence is shown in our experimental result in Table 1 of the paper. Through our novel hierarchical searching strategy coupled with the epistemic uncertainty, during inference step, we can effectively mask out those false positive predictive bounding boxes even if they have a high confidence. We note that redundant small bounding phenomenon is a common problem in dense object detection that challenges most existing works (e.g., Li et al., Generalized focal loss, NeurIPS 2020).
>
> **Q3: Clarification on use of masks to generate final bounding boxes**
>
> Please refer to the answer to **Q2** in the General Response.
>
> **Q4: How does the binary RL mask help reduce unnecessary bounding boxes in inference step**
>
> Please refer to the answer to **Q4** in the General Response.

---

> > ### Author Response · Authors · 2024-08-12
> >
> > Dear Reviewer 4PiS,
> >
> >
> > Thank you again for your constructive comments and insightful questions! In the rebuttal, we have provided
> >
> > - a clear definition of false positive cases with additional examples (please refer to Figure 1 in the attached PDF),
> >
> > - justification on why simple post-processing may not be sufficient to remove redundant bounding boxes,
> >
> > - clarification on use of masks to generate final bounding boxes (please refer to the answer to Q2 in the General Response),
> >
> > - justification on how the binary RL mask reduces unnecessary bounding boxes (please refer to the answer to Q4 in the General Response).
> >
> > By addressing your comments, we believe that quality of the paper has been improved and we appreciate the reviewer's support on that. We are more than happy to answer any additional questions that you may have.

---

> > > ### Comment · Reviewer_4PiS · 2024-08-14
> > >
> > > The authors have addressed all my questions. The paper is deserving of acceptance in this venue. Despite my limited expertise in dense object detection, I find it really accessible and comprehensible.

---

> > > > ### Author Response · Authors · 2024-08-14
> > > >
> > > > Many thanks for reading our rebuttal and confirming that it has addressed all your concerns. We appreciate your support for the acceptance of the paper!

---

### Author Rebuttal · Authors · 2024-08-07

**General Response**

We would like to thank all the reviewers for their constructive suggestions and comments. Here, we summarize our responses to some common questions raised by multiple reviewers:

**Q1: Poor performance of AIRS in MS COCO Large Objects (Reviewers mhHL and Q21q)**

In this work, we aim to improve the detection performance by having a good balance between objects of different sizes and the $AP$ metric is designed to assess the overall effectiveness in terms of detecting objects in all granularities. Compared to competitive baselines, AIRS is superior on all datasets. We agree that placing more focus on smaller and more difficult objects lowers the performance of AIRS on $AP^{L}$ and $AP^{M}$ in MS COCO. However, this is an expected behavior as MS COCO has most of the objects being very large and therefore, the cost of missing smaller objects in the existing two-stage detectors seem to be very low. As  such, many two-stage detectors have superior performance (see Table 1 of the paper). In contrast, as our technique leverages a one-stage detector to better cover dense objects, it is relatively less effective to detect very large objects (which is evidenced by the lower performance by all one-stage detectors in Table 1). It is worth mentioning that in other datasets, AIRS outperforms all baselines even on the large objects. In the case of Pascal VOC 2012, it is relatively easier and does not contain very large objects. As such, one-stage detectors perform comparable or even better than the two-stage detectors.  As for Open Image V4, despite being challenging, it contains a good amount of training samples with larger objects, which provides enough supervision for models to detect these large objects. As such, all single-detectors including our technique perform comparable or even better compared to two-stage detectors.


**Q2: Clarification on use of masks to generate final bounding boxes (Reviewers 4PiS and Q21q)**

We run the trained RL agent on the test image's FPN to generate RL masks. Based on the masked evidential Q-value estimate, the agent selects the next action, which would be either a downward or upward movement. Then, the agent moves to the next patch and continues the process until receiving an upward movement in layer $L$ or reaches the maximum time step i.e. $T$. After the hierarchical searching process, we can have a binary RL mask by recording which patches are visited by RL agent through it actions (denoted as 1 in RL mask), and which are not visited by the RL agent (denoted as 0 in RL mask). Given the RL mask, which is a three-level binary mask covering the feature pyramid network (FPN), each pixel in the FPN will be assigned a confidence score in the quality evaluation branch to decide if it is a positive anchor or not by (comparing with a threshold). Those pixels covered by zero RL masks will have their confidence score reset by 0, and other pixels will maintain the same confidence score. In that way, RL masks serve as an additional filter to further eliminate the "false positive" predicted bounding boxes. We show an illustration of RL masks in Figure 1 of the attached PDF.


**Q3: Comparison with two stage detectors like RPN (Reviewer 8nSu)**

There are key differences between two stage detectors and AIRS.
The former usually relies on a Region Proposal Network (RPN), which is less effective to capture all targeted objects especially in a dense scenario. This is because, RPN selects anchors from the candidate anchors provided by the region proposal network based on the confidence score resulting into missing many true positive object anchors with a low confidence. In contrast, FPN in AIRS is based on multi-scale feature representations. Thus, the number of selected anchors in all layers is far more than the ones proposed by RPN and therefore avoiding the missing of important object anchors. To tackle the many false positive anchors in the FPN based approaches, we propose a novel hierarchical search coupled with an effective exploration-exploitation strategy leveraging evidential Q-learning. As a result, AIRS effectively removes the false positive bounding boxes without removing the less confident true positive objects. This phenomenon is also demonstrated in Table 1 of the paper, where two-stage detectors result in a lower performance compared to AIRS in dense object detection.


**Q4: How does the binary RL mask help reduce unnecessary bounding boxes in inference step (Reviewers 4Pis, mhHL, and Q21q)**

We are handling two different types of false positive bounding boxes. The first category involves bounding boxes that capture only background with no targeted object. To remove those patches, our novel exploration-exploitation strategy plays a major role. Specifically, during the adaptive hierarchical search in the top-down fashion, exploration of the higher layer quickly discovers that there is no object in the lower-level granularity. Specifically, both Q-value as well  as epistemic uncertainty (see Eq. 4) remains low, leading to removing  bounding boxes on backgrounds. The second category involves bounding boxes that cover only a part of a given object and are embedded in the larger bounding box that covers the whole object. As our approach works in the top-down fashion, once the RL agent explores the bigger bounding box covering the full object, the model assigns a very low epistemic uncertainty for partially covering bounding boxes. As such, the model avoids going downward in a lower level granularity in the action space resulting removing unnecessary partially covered bounding boxes.

---

### Decision · Program_Chairs · 2024-09-25

**Decision:**

Accept (poster)

**Comment:**

This paper presents a novel approach to dense object detection called Adaptive Important Region Selection (AIRS). The method utilizes evidential Q-learning and a hierarchical search strategy to identify important regions in images, aiming to reduce false positive detections while improving performance on small object detection.

Initially, the paper garnered mixed reviews, with reviewers acknowledging its innovative aspects but also raising concerns about certain claims and comparisons, e.g., poor performance on large objects, comparison with two-stage detectors like RPN. After the rebuttal stage, most of the concerns have been addressed, but the claim between AIRS and RPN still deserves further clarification, as commented by Reviewer 8nSu.

Overall, the paper presents an interesting and potentially impactful approach to dense object detection. The overall consensus is that the work is technically sound and makes valuable contributions to the field.

The AC is happy to recommend the paper for acceptance. Congratulations! Please be aware that the authors are still strongly recommended to incorporate the response in the rebuttal and adequately address the raised issues in the final camera-ready version.